# FUT6 deficiency compromises basophil function by selectively abrogating their sialyl-Lewis x expression

Kia Joo Puan [1,9,10], Boris San Luis[1,10], Nurhashikin Yusof[1], Dilip Kumar [1], Anand Kumar Andiappan [1], Wendy Lee[1], Samanta Cajic [2], Dragana Vuckovic[3], Jing De Chan [1], Tobias Döllner[1], Han Wei Hou [4,5], Yunxuan Jiang[6], Chao Tian[6], the 23andMe Research Team*, Erdmann Rapp [2,7], Michael Poidinger [1], De Yun Wang[8], Nicole Soranzo [3], Bernett Lee[1] & Olaf Rötzschke [1✉]

Sialyl-Lewis x (sLe$^x$, CD15s) is a tetra-saccharide on the surface of leukocytes required for E-selectin-mediated rolling, a prerequisite for leukocytes to migrate out of the blood vessels. Here we show using flow cytometry that sLe$^x$ expression on basophils and mast cell progenitors depends on fucosyltransferase 6 (FUT6). Using genetic association data analysis and qPCR, the cell type-specific defect was associated with single nucleotide polymorphisms (SNPs) in the FUT6 gene region (tagged by rs17855739 and rs778798), affecting coding sequence and/or expression level of the mRNA. Heterozygous individuals with one functional FUT6 gene harbor a mixed population of sLe$^{x+}$ and sLe$^{x-}$ basophils, a phenomenon caused by random monoallelic expression (RME). Microfluidic assay demonstrated FUT6-deficient basophils rolling on E-selectin is severely impaired. FUT6 null alleles carriers exhibit elevated blood basophil counts and a reduced itch sensitivity against insect bites. FUT6-deficiency thus dampens the basophil-mediated allergic response in the periphery, evident also in lower IgE titers and reduced eosinophil counts.

[1] Singapore Immunology Network (SIgN), A*STAR (Agency for Science, Technology and Research), Singapore, Singapore. [2] Max Planck Institute for Dynamics of Complex Technical Systems, Magdeburg, Germany. [3] Department of Human Genetics, Wellcome Sanger Institute, Cambridge, UK. [4] Lee Kong Chian School of Medicine, Nanyang Technological University, Singapore, Singapore. [5] School of Mechanical and Aerospace Engineering, Nanyang Technological University, Singapore, Singapore. [6] 23andMe, Inc., Sunnyvale, CA, USA. [7] glyXera GmbH, Magdeburg, Germany. [8] Department of Otolaryngology, National University of Singapore, Singapore, Singapore. [9] Present address: Shanghai Junshi Biosciences Co., Ltd, Shanghai, China. [10] These authors contributed equally: Kia Joo Puan, Boris San Luis. *A list of authors and their affiliations appears at the end of the paper. ✉email: olaf_rotzschke@immunol.a-star.edu.sg

Sialyl Lewis x (sLe$^x$, CD15s) is a tetra-saccharide composed of N-acetylneuraminic acid (sialic acid), galactose, fucose, and N-acetyl glucosamine (NeuNAcα2,3Galβ1,4(Fucα1,3)-GlcNAc). sLe$^x$ expression has been reported on various leukocytes, including monocytes, neutrophils, activated T cells[1], and regulatory T cells[2]. It forms the terminal group of O-glycans on molecules such as PSGL-1, CD43, or CD44 but is also found on N-glycans of some proteins. Displayed on glycolipids of erythrocytes, it is one of the established blood group antigens. On leukocytes, sLe$^x$ plays a crucial role in extravasation. Their transient interaction with E-selectin on activated epithelial surface of blood vessels[3] results in the deceleration of the cell in the blood stream ("rolling"), which allows a firm integrin-dependent adhesion to facilitate trans-endothelial migration[4]. In a recent study, removal of sialic acid from immunoglobulin E (IgE) has been found to attenuate the allergic pathogenicity of IgE on basophils and mast cells[5].

sLe$^x$ is generated in the Golgi compartment by a complex series of enzymatic reactions. The rate-limiting step in this process is the addition of fucose by an α-1,3-fucosyltransferase (FUT). Of the six variants encoded by the human genome, FUT6 and FUT7 seem to be specifically involved in the generation of E-selectin ligands. FUT7 is reportedly the dominant transferase for leukocytes[1,6], whereas FUT6 is only known to be involved in the sLe$^x$ production on plasma proteins and epithelial cancer cells[7,8]. While the disruption of the entire fucosylation pathway by a defective GDP-fucose transporter (SLC35C1) results in very severe symptoms (leukocyte adhesion deficiency II)[9], genetic defects in FUT6 or FUT7 are apparently without any apparent clinical consequences[7,10]. A functional redundancy within the FUT gene family as well as the presence of structural paralogs of the sLe$^x$ glycan seems to prevent the manifestation of deleterious effects. In the case of FUT7, a natural mutation of the gene was reported that completely abrogated the sLe$^x$ expression on neutrophils[10]. The loss, however, is compensated by CD65s (VIM-2), a functional sLe$^x$ paralog generated by FUT4.

In evolutionary terms, FUT6 is a much younger gene than FUT7. While the latter is also present in rodents, FUT6 was generated together with its orthologs FUT3 and FUT5 in primates by a gene duplication event. FUT7 is a dominant gene for synthesis of sLe$^x$ in leukocytes. In contrast, FUT6 is known as a "plasma-type" FUT making sLe$^x$ on plasma proteins[7,11] and epithelial cancer cell lines[12]. In a recent study, patients who have high expression of FUT3, FUT6, and FUT7 showed adverse effects on event-free survival and overall survival after receiving chemotherapy[13]. Aberrant FUT6 expression has been associated with the metastasis of gastrointestinal cancer[14]. However, a study of a small Javanese cohort revealed that about 10% of these individuals had no functional FUT6 genes suggesting that a loss of function seems to be without major consequences[7]. Although this study gained "fucosyltransferase 6 deficiency" (FUT6 deficiency) an entry in the "Online Mendelian Inheritance in Man" registry (OMIM #613852), it has not been linked yet to any clinical condition or syndrome.

In this study, we could show now that defects in the FUT6 gene have indeed functional implications. Based on the data derived from a deeply characterized Singapore Systems Immunology Cohort (SSIC), we could establish that single-nucleotide polymorphisms (SNPs) in the FUT6 gene region selectively compromise the sLe$^x$ expression in basophils and mast cell progenitors (pMCs). Moreover, at least in basophils the expression of FUT6 was found to be controlled by random monoallelic expression (RME). sLe$^{x-}$ basophils are therefore present not only in FUT6 −/− individuals but also in FUT6 −/+, where they make up about 50% of the entire basophil pool. Flow chamber experiments confirmed that sLe$^{x-}$ basophils fail to roll on E-selectin-coated surfaces. In line with this

migratory defect, blood counts of >400,000 genotyped British donors revealed elevated basophil numbers for carriers of FUT6 null alleles. Questionnaire data from a large 23andMe cohort further indicates reduced sensitivity to insect bites, which, together with lower IgE titers observed for the Singapore cohort, points to a dampening effect of the FUT6 deficiency on the IgE-mediated allergic response.

## Results

**Variable sLe$^x$ expression on basophils**. On the cell surface, sLe$^x$ (Supplementary Fig. 1a, b) is usually detected by antibodies specific for CD15s. The marker is reportedly absent on mast cells[15] and eosinophils[16] but expressed by neutrophils, monocytes[1] and basophils[17]. When analyzing whole-blood samples from our SSIC cohort by fluorescence-activated cell sorting (FACS), we indeed observed a clear positive staining on neutrophils and monocytes while eosinophils stained negative (Fig. 1a). The CD15s staining of basophils, however, diverged from the earlier study. While the majority of the cohort samples showed the reported positive staining (upper panel), in some individuals the CD15s expression by basophils was completely absent (lower panel). Moreover, in some donors the CD15s expression was even bimodal (CD15s$^{bimodal}$), indicating the co-existence of two distinct populations CD15s$^{low}$ and CD15s$^{high}$ basophils (Fig. 1b). Although some variation was observed among CD15s$^{bimodal}$ individuals with regard to the relative peak heights, the pattern remained stable over time. Donors could thus be divided into three categories: CD15s$^{low}$ (no or low sLe$^x$ expression), CD15s$^{bimodal}$ (two distinctly separated peaks), and CD15s$^{high}$ (high sLe$^x$ expression).

The phenomenon was strictly cell-type specific as other leukocytes, such as monocytes and neutrophils, showed normal CD15s staining (Fig. 1c). It was observed only for the sialylated form of the Lewis antigen (CD15s) and not for the non-sialylated version (CD15), which at lower levels was also expressed on basophils. The differences in the variation of CD15s are particularly evident in the boxplots of the monocyte and basophil populations of the cohort (Fig. 1d).

**Basophil rolling depends on sLe$^x$ expression**. In order to determine whether the variations in sLe$^x$ expression translate into any functional defects, we tested the capacity of basophils to roll on E selectin-coated surfaces (Fig. 2a). For this experiment, basophils isolated from CD15s$^{high}$, CD15s$^{bimodal}$, and CD15s$^{low}$ individuals were passed through a flow chamber coated before with recombinant E-selectin. Their interaction with the inner wall was recorded by video imaging (Supplementary Movie 1). As expected, the cells from CD15s$^{high}$ individuals showed clear rolling while cells from CD15s$^{low}$ individuals completely failed to interact with the surface. In line with the relative fraction of CD15s$^{high}$ basophils, the number of rolling cells from CD15s$^{bimodal}$ individuals was roughly halved, suggesting that also in these individuals the subset of CD15s$^{low}$ cells fails to interact with E-selectin (Fig. 2b). Thus, for basophils the rolling on E-selectin strictly depends on the presence of sLe$^x$.

**sLe$^x$ expression on basophils is controlled by FUT6**. In order to identify the protein causing the basophil-specific sLe$^x$ deficiency, we carried out RNA-sequencing (RNA-seq) of FACS-sorted basophils. To avoid "noise" due to inter-individual variations in gene expression, we isolated CD15s$^{high}$ and CD15s$^{low}$ fractions of basophils from CD15s$^{bimodal}$ donors and carried out a paired analysis of the obtained RNA-seq data (Fig. 3a). Surprisingly, the analysis revealed only a single candidate, FUT6 (Fig. 3b). The volcano plot indicated more than sixfold upregulation of the

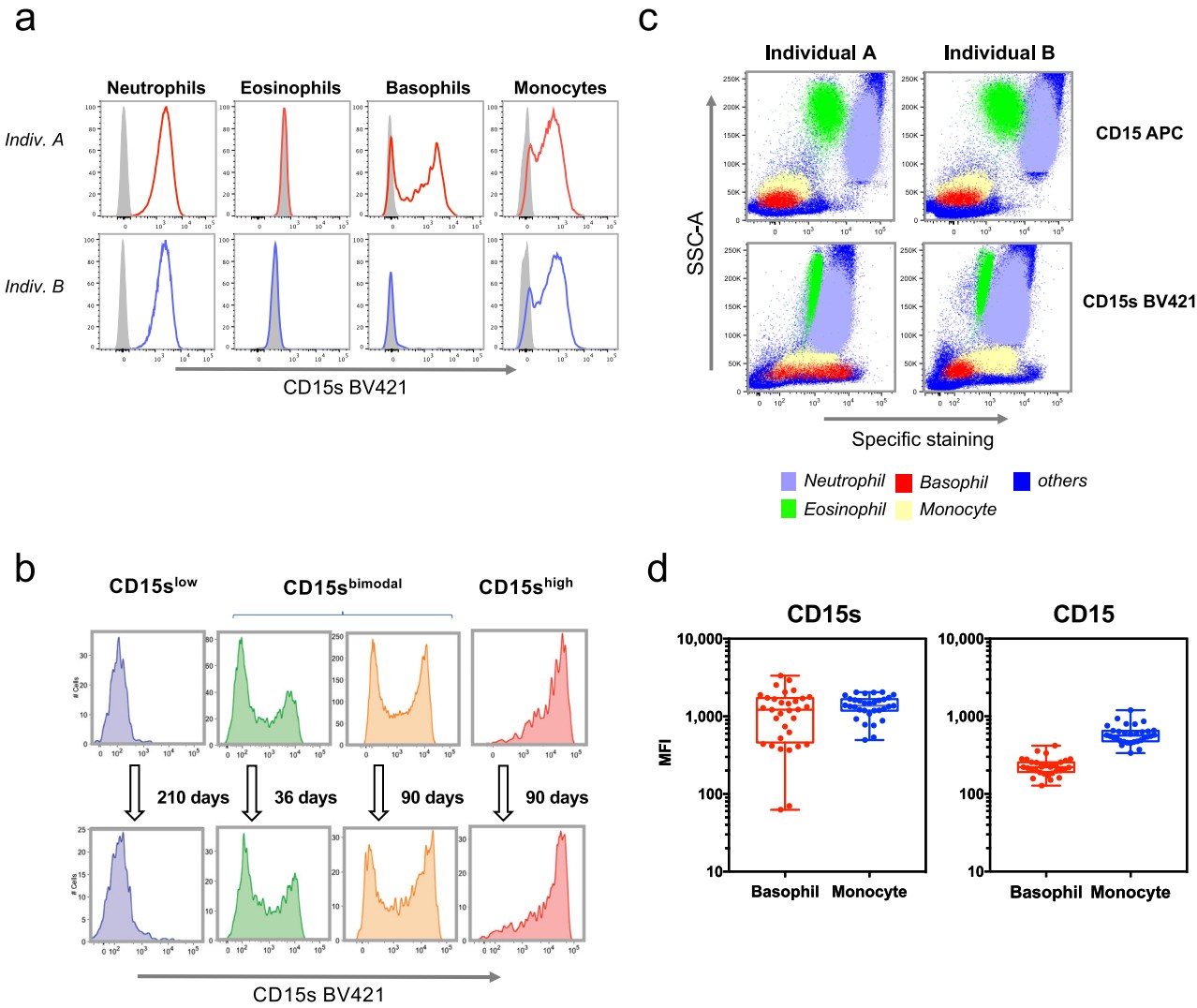

**Fig. 1 CD15s (sLe$^x$) expression on human basophils. a** CD15s staining. FACS analysis of whole-blood samples of two individuals differing in the CD15s expression on their basophils. CD15s staining is shown for neutrophils, eosinophils, basophils, and monocytes. Gray peaks represent the background staining based on "fluorescence minus one" (FMO). **b** Temporal stability of the CD15s pattern. The CD15s staining on basophils is shown for four individuals with either low (CD15s$^{low}$), bimodal (CD15s$^{bimodal}$), or high CD15s expression on basophils (CD15s$^{high}$). Two examples of bimodal expression are shown differing in the ratio between the CD15s$^{low}$ and the CD15s$^{high}$ peak. The FACS plots indicate CD15s measurement from four different individuals between two time points. **c** Comparison of CD15 and CD15s expression on various leukocytes. The dot plots represent FACS profile of whole-blood samples of two individuals stained with anti-CD15 and anti-CD15s. The specific staining is shown in reference to the side scatter (SSC-A); the location of neutrophils (violet), eosinophils (green), basophils (red), monocyte (yellow), and other cells (blue) is indicated. Cell populations were defined by cell type-specific gating (Supplementary Fig. 4). **d** Cohort-wide distribution of CD15s and CD15 expression. The boxplots indicate the mean fluorescence intensity (MFI) of CD15s ($n = 32$) and CD15 ($n = 29$) on basophils and monocytes. Each dot represents one individual; all samples were obtained from healthy volunteers of the Singapore SSIC cohort.

*FUT6* mRNA levels in CD15s$^{high}$ compared to CD15s$^{low}$ fractions ($p$ value $<10^{-12}$). Besides *FUT6*, no other genes or gene signatures were evident that discriminated CD15s$^{high}$ from CD15s$^{low}$ basophils. This applied also for the RNA levels of *FUT7*, the other enzyme involved in the sLe$^x$ synthesis, which were equally high in both fractions. The downregulation of FUT6 in CD15s$^{low}$ basophils was confirmed by quantitative PCR (qPCR; Fig. 3c). In line with the RNA-seq results, the average mRNA levels encoded by the gene were about sixfold higher in CD15s$^{high}$ fractions compared to the corresponding CD15s$^{low}$ fractions ($p = 0.0002$). No difference was observed for *FUT7* but the amount of *FUT7* RNA in both fractions was comparable to that of *FUT6* in CD15s$^{high}$ cells. Thus, despite its presence in basophils and its demonstrated role in neutrophils[6], *FUT7* does not seem to contribute to the sLe$^x$ generation in basophils.

A very similar result was obtained when analyzing the published RNA-seq data from Monaco et al.[18] (Supplementary Fig. 2). Also, in this study the amount of *FUT6* and *FUT7* RNA in basophils was roughly comparable. More importantly, while *FUT7* was expressed in every cell analyzed, *FUT6* was detected at substantial amounts only in basophils and progenitor cells, a finding in line with the suggested cell-type specific of the phenotype variations of CD15s. When analyzing the CD34+ CD117+ progenitor pool of the peripheral blood mononuclear cell (PBMC) compartment by FACS, we noted that only FcεRI+ pMCs[19] exhibit the same CD15s variation as basophils, pMCs are a tiny population comprising only approximately mean 0.003% of the PBMC (range 0.002–0.004%), and the phenotype was not shared by the bulk of FcεRI− progenitors (Supplementary Fig. 3). Both basophils and pMCs are closely related and derived from the

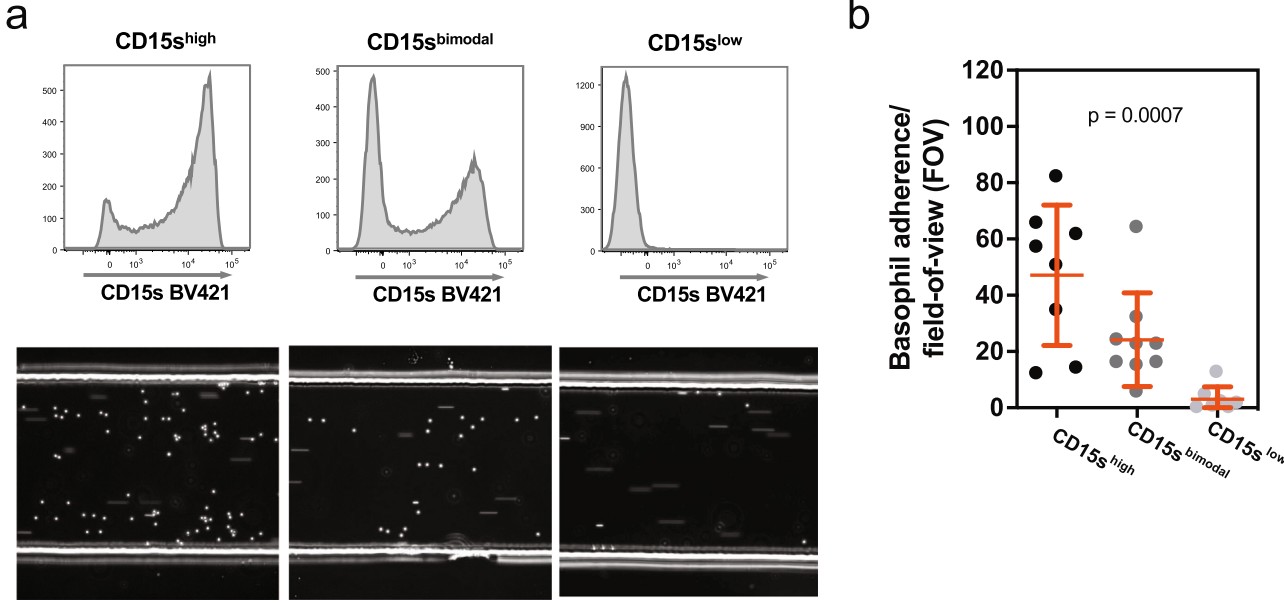

**Fig. 2 Rolling of basophils on E-selectin-coated surfaces. a** Flow chamber experiment with basophils from CD15s[high], CD15s[bimodal], and CD15s[low] individuals. Basophils isolated by negative selection were flushed through E-selectin-coated flow chambers. The upper panel shows the CD15s FACS profile of the isolated basophils, the lower panel a snapshot taken during their passage through the flow chamber. The dots represent adherent cells rolling on the surface, the streaks non-adherent cells carried by the stream; videos can be downloaded as Supplementary Movie 1. **b** Statistical analysis. The dots indicate the number of adherent basophils per field of view (FOV). Each dot represents an independent frame. The number of adherent basophils were found to be significantly different between the basophil subsets via Kruskal–Wallis test ($p = 0.0007$).

same myeloid progenitor population[20]. Notably, while neutrophils lack any detectable expression of *FUT6* RNA, their levels of *FUT7* were >30-fold higher compared to basophils, explaining the tight link between CD15s and *FUT7* for these cells (Supplementary Fig. 2).

**sLe^x expression in basophils is controlled by *FUT6* SNPs.** Although many genes are involved in the synthesis of sLe^x (Supplementary Fig. 1), the mRNA analysis revealed only *FUT6* as the cause for the variations in basophils (Fig. 1b). This was further confirmed when we analyzed our SSIC database for sLe^x-related quantitative trait loci (pQTL). While the genotypes had been determined before at the genome-wide level with microarray chips[21,22], the CD15s expression on basophils still needed to be defined. For this, we carried out a FACS analysis of cryopreserved PBMC samples using a panel that allowed to determine the sLe^x expression on the various cell subsets (Supplementary Fig. 4).

In total, 32 SNPs associated with the mean fluorescence intensity (MFI) of the CD15s expression of basophils at an adjusted $p$ value <0.05 (Supplementary Data 1). All of them were located within or proximal to the *FUT6* locus (Fig. 4a). A linkage analysis revealed that the strongest associations were detected for two linkage blocks (Fig. 4b and Supplementary Fig. 6). On top of the list was a linkage block with nearly perfect linkage ($r^2 > 0.95$) that was tagged by rs778798 (termed "rs778798-LB"). It comprised six non-coding SNPs and associated with the CD15s MFI with $p = 2.8 \times 10^{-15}$ (Fig. 4c). The second linkage block (termed here as "rs17855739-LB") correlated with the CD15s expression with slightly higher $p$ values. It comprised nine SNPs of which the tag SNP rs17855739 was representing a coding SNP that had been previously associated with FUT6 deficiency[8] (Supplementary Data 1 and Fig. 4b). The SNP is located in the catalytic domain of *FUT6* where it encodes for the E_{247}→K substitution and associated with the CD15s expression on

basophils with $p = 1.9 \times 10^{-10}$ (Fig. 4c). Another coding SNP (rs145035679) linked to rs17855739 with $r^2 = 0.77$ associated with CD15s expression with a slightly higher $p$ value ($3.34 \times 10^{-06}$). It encodes for a stop codon that truncates the *FUT6* gene product by 45 amino acids (Y_{315}→stop)[7].

The $r^2$ value between rs778798 and rs17855739 was <0.05, suggesting that each of the two linkage blocks had an independent effect on the basophil-specific CD15s expression. This assumption was indeed confirmed after carrying out a conditional SNP analysis (Supplementary Fig. 7a). The genotype/phenotype correlation for the two tag SNPs rs778798 and rs17855739 revealed that in both linkage blocks the low CD15s expression associated with the minor allele (A for rs778798 and T for rs17855739). For rs778798, the MFI of basophil CD15s decreased from 1629 (CC) over 602.7 (AC) to 57.6 (AA) (Fig. 4c, left panel), while for rs17855739 the highest MFI value was 1496 (CC) followed by 776.4 (CT) and 269.7 (TT) (Fig. 4c, middle panel). Likewise, for rs145035679 the MFI decreased from GG to GT but the imputation was only of moderate confidence (0.5 > info score < 0.9) (Fig. 4c, right panel).

**Impact of *FUT6* null alleles on the bimodal sLe^x expression pattern.** *FUT6* does not appear to be under lineage constraint as detectable in gnomAD (https://gnomad.broadinstitute.org/) and for both linkage blocks the ancestral allele is represented by the major allele. As the D' value of the genetic linkage between the two tag SNPs is 1, the deleterious variants associated with the derived alleles of these SNPs had likely been introduced independently of each other into the ancestral *FUT6* sequence. When inspecting the genotype combinations of rs778798 and rs17855739 in our cohort, we noted that AA/TT, AA/CT, and AC/TT were indeed missing (inset in Fig. 5a). This same trend is also evident in two larger but ethnically more diverse cohorts from the UK Biobank and INTERVAL study and 23andMe.

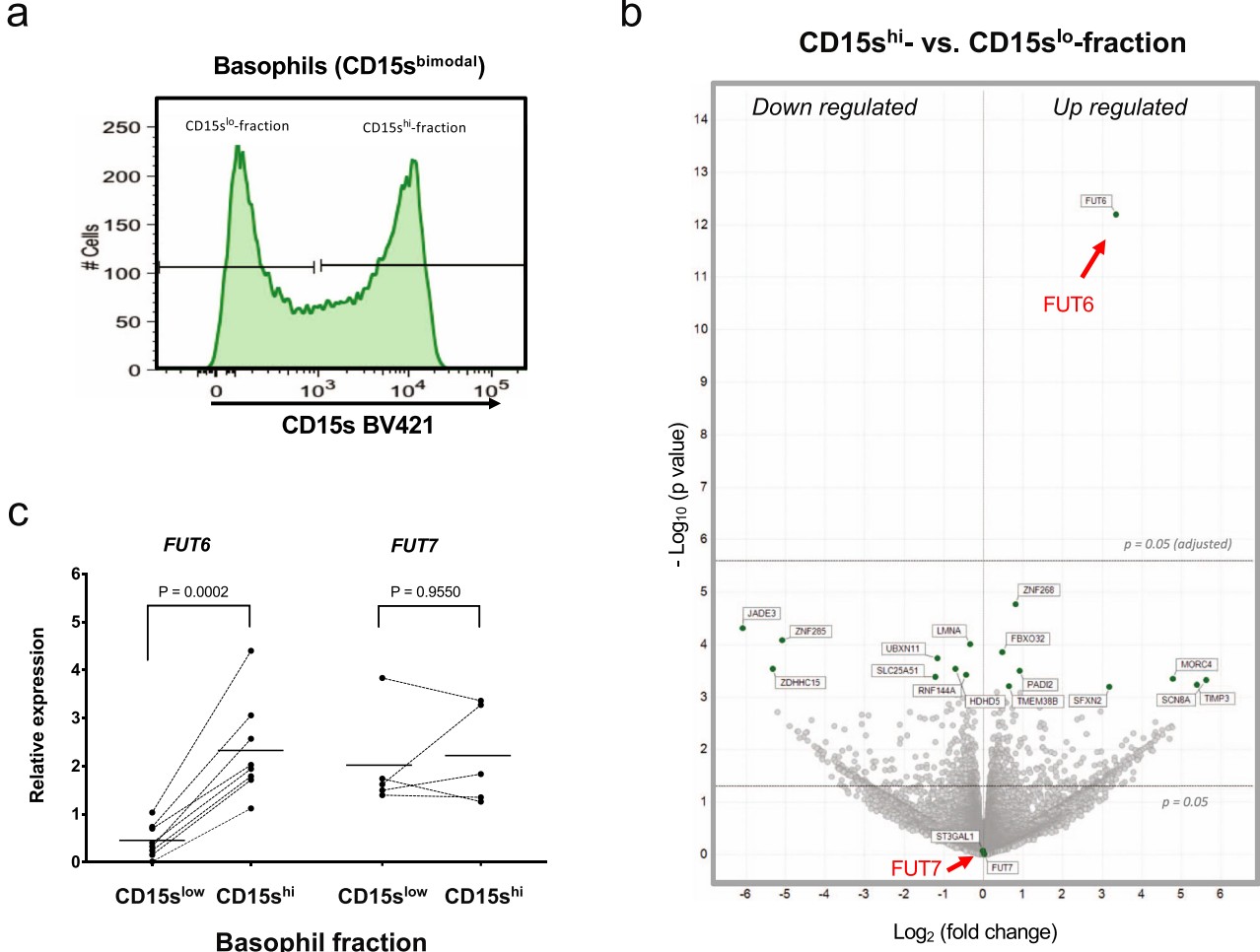

**Fig. 3 Gene expression analysis of CD15s^high and CD15s^low basophils. a** FACS sorting of CD15s^high and CD15s^low fraction of CD15s^bimodal individuals. FACS profile and sorting gates of the CD15s^low and CD15s^high basophil fraction are shown for one representative donor from six donors. **b** Volcano plot. The plot summarized the edgeR paired RNA-seq results of the pairs of CD15s^high vs. CD15s^low fractions obtained from six CD15s^bimodal donors. Each dot represents a gene and the plot displays the fold change (log2) vs. nominal $p$ value ($-\log10$). Boundaries for nominal and adjusted $p$ value are indicated. **c** Validation by qPCR. The mRNA expression of *FUT6* ($n = 8$) and *FUT7* ($n = 6$) in the CD15s^low and CD15s^high fraction validated by qPCR. Paired samples are connected by lines; $p$ values are indicated. Statistical comparison was performed using Wilcoxon matched-pairs signed-rank test.

While both cohorts comprised around 500,000 individuals, AA/TT was completely absent, which is reflected in $D'$ values of 0.99 for both cohorts (Supplementary Fig. 7b). Thus, a haplotype combining the derived alleles of both linkage blocks on one strand virtually does not exist.

As the null alleles encoded by the two linkage blocks are mutually exclusive and the functional implications of the associated *FUT6* defect are likely to be comparable, we combined them in an additive fashion (Fig. 5a). In total, we identified two homozygous *FUT6* −/− individuals in our cohort where both null alleles where encoded by rs778798-LB (A̲A̲/CC), two individuals with two null alleles encoded by rs17855739-LB (CC/T̲T̲), and additional 8 individuals, in which each of the two linkage blocks contributed one of the two defective *FUT6* genes (A̲C/C̲T̲) (Fig. 5a, left panel). Accordingly, *FUT6* +/− individuals were represented by AC/CC and CC/CT (each containing only a single null allele) while *FUT6* +/+ corresponded to CC/CC (no null alleles). When using this assignment for genotype/phenotype correlations, the average CD15s MFI on basophils of increased from 176.7 on *FUT6* −/− individuals to 788.3 for *FUT6* +/− and to 1852 for *FUT6* +/+ individuals (Fig. 5a, right panel). Compared to the single SNPs (Fig. 4c), the $p$ value for the association with the *FUT6* state further decreased to $2.09 \times 10^{-38}$ (Fig. 5a, right panel).

Cohort-wide correlation of the three *FUT6* states with the CD15s expression on basophils confirmed the expected association between the number of *FUT6* null alleles and the absolute amount of sLe^x on the surface of these cells (Fig. 5b). Moreover, when using the percentage of CD15s^high cells of the total basophil population as a measure of the basophil phenotype CD15s^low, CD15s^bimodal, and CD15s^high, we noted that *FUT6* −/− individuals were mostly of the CD15s^low type (i.e., <10% of the basophils expressing CD15s), while most of the *FUT6* +/+ individuals showed the largely unimodal expression characteristic of the CD15s^high type. Importantly, most of the *FUT6* −/+ individuals were of CD15s^bimodal type harboring a mixed population of CD15^high and CD15s^low cells. The partial overlap between the *FUT6* +/− and −/− subsets suggests, however, the presence of additional null alleles not encoded by the two linkage blocks.

**RME of *FUT6*.** The heterozygous expression of a gene pair with one null allele typically results in an intermediate expression level by the respective cells rather than a split of the cell population into a positive and negative subset. One mechanism that can drive this unusual segregation is RME[23]. RME refers to the stochastic transcriptional inactivation of one of the two existing autosomal

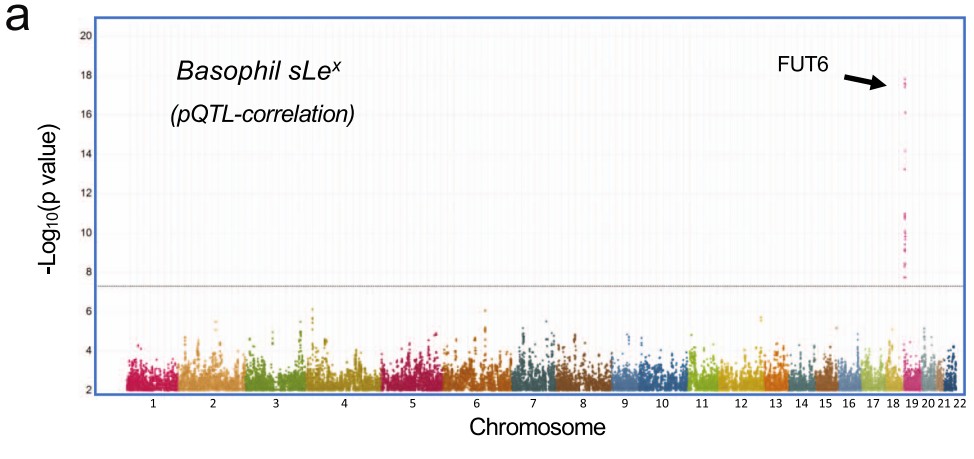

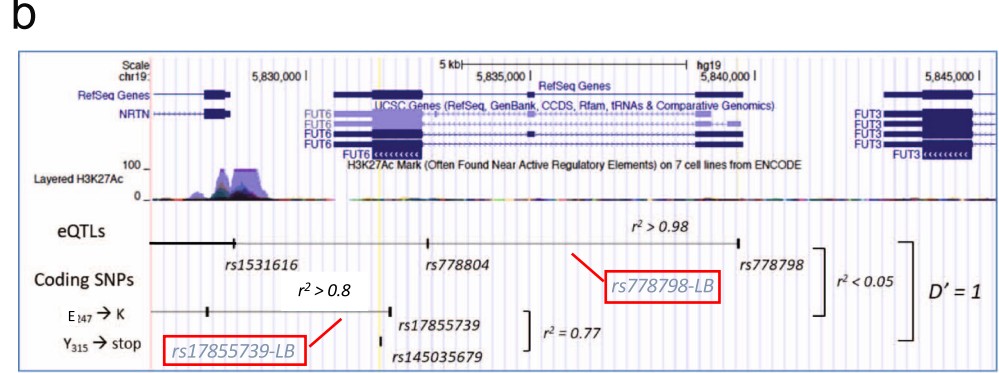

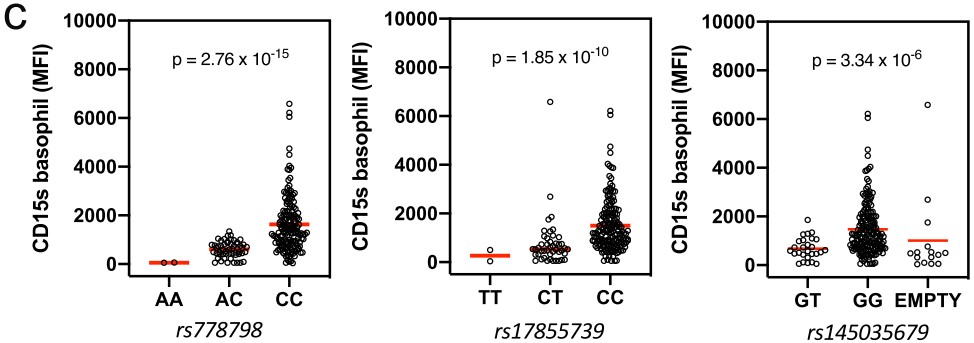

**Fig. 4 Genetic control of sLe$^x$ expression on basophils. a** Genome-wide pQTL correlation. PBMC samples of genotyped individuals ($n = 229$) were analyzed by FACS. The Manhattan plot summarizes the association of the CD15s MFI of the basophils with the genotypes of the 2 million SNPs covered by this analysis. The $x$-axis represents the location of SNPs on the 22 autosomal chromosomes, the $Y$-axis the nominal $p$ values ($-$log10) of their association with CD15s. A clear association was observed for SNPs on chromosome 19, which are all located proximal to the *FUT6* gene. **b** *FUT6* gene locus. The upper part of the figure displays the location of intron and exon elements of *FUT6* and neighboring genes together with the density of H3K27Ac marks. The lower part indicates the location of some of the top SNPs identified in the Manhattan plot. They are arranged into two independent linkage blocks rs778798-LB and rs17855739-LB ($r^2 < 0.05$, $D' = 1$). rs778804-LB comprises only of non-coding SNPs (rs778798, rs778804, rs1531616, rs1678852, rs199921063, and rs3763045) while rs17855739-LB has also two coding SNPs (rs17855739 and rs145035679). Closely linked SNPs are connected by a horizontal line; the genetic linkage is indicated ($r^2$ and $D'$); complete list of SNPs is displayed in Supplementary Data 1. **c** Genotype/phenotype association. The dot plots show the association of the CD15s MFI on basophils with the genotypes of rs778798 AA ($n = 2$), AC ($n = 53$), CC ($n = 164$), and the two coding SNPs rs17855739 (E$_{247}$→K) TT ($n = 2$), CT ($n = 45$), CC ($n = 173$), and rs145035679 (Y$_{315}$→stop) GT ($n = 27$), GG ($n = 183$), EMPTY ($n = 15$). Each dot represents one individual, $p$ values are indicated. Plots were generated with the data collected for individuals of Chinese ethnicity from the SSIC cohort[21,22]. Significance was determined by Kruskal–Wallis tests.

genes. It is a rare and cell type-specific phenomenon observed in about 3% of the genes. The choice of the inactivated allele is random, but if one of the two alleles is defective, on the cell population level, it results in a striking bimodal expression pattern consistent with the observed staining in CD15s$^{bimodal}$

individuals. To confirm that the transcription of *FUT6* is actually under RME control, we used heterozygous alleles of coding SNPs to tag each of the two corresponding *FUT6* transcripts of the maternal and paternal genes in CD15s$^{bimodal}$ individuals. In the case of RME, the sequence comparison of mRNAs isolated from

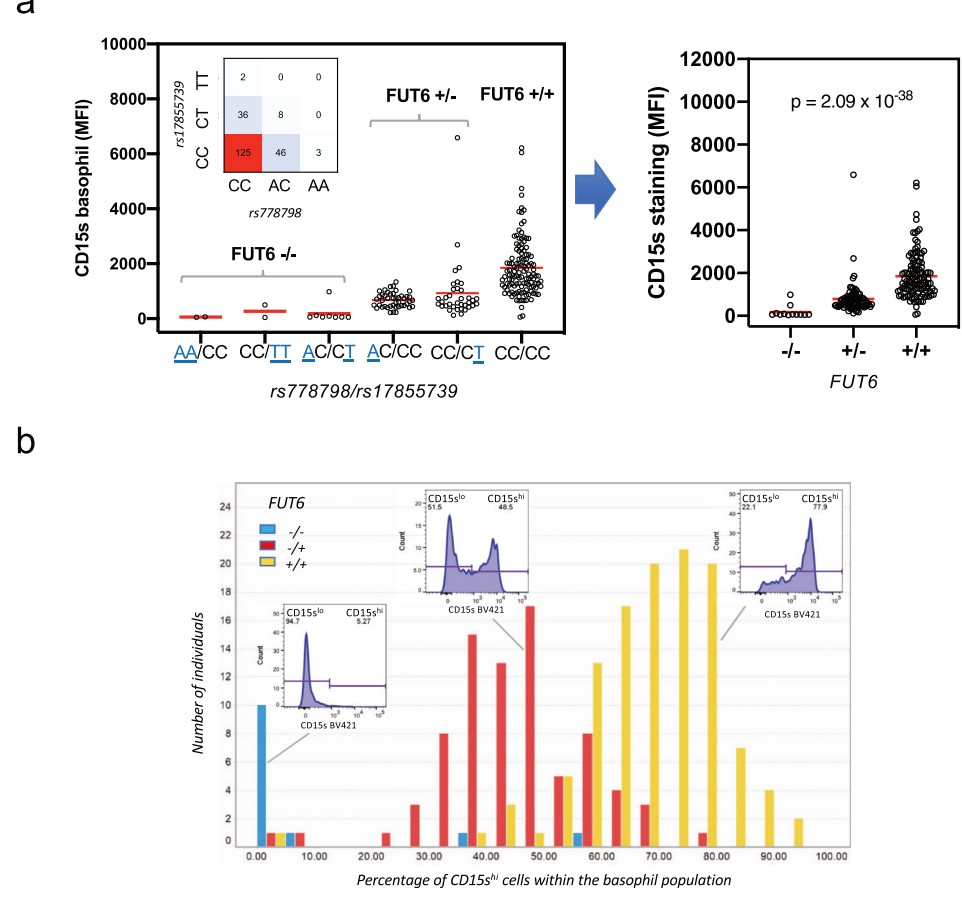

**Fig. 5 Association of *FUT6* null alleles with the bimodal sLe^x expression in basophils. a** rs778798/rs17855739-defined *FUT6* states. The expression states of *FUT6* can be imputed from genotypes of rs778798 and rs17855739. The left panel displays the mean fluorescence values (MFI) of the CD15s staining of basophils in reference to the rs778798/rs17855739 genotype combination (null alleles are underlined; the inset shows the number of individuals with all genotype combinations detected in the SSIC cohort). The right panel shows the MFI of the CD15s staining vs. the imputed *FUT6* status (+/+, +/−, −/−). Significance was determined by Kruskal–Wallis tests. **b** Impact of FUT6 status on the sLe^x expression on basophils. The type of CD15s basophil expression by a donor (CD15s^low, CD15s^bimodal, CD15s^high) can be described by the percentage of CD15s^high cells within the basophil population. The histogram shows the distribution of this parameter within the SSIC cohort. The data were grouped based on the imputed *FUT6* status (blue: −/− (n = 12); red: +/− (n = 81); yellow: −/− (n = 122)). Insets show representative FACS histograms of the CD15s basophil staining for each *FUT6* status. Bars are binned over a range of 5%.

the CD15s^high and CD15s^low basophil fraction should reveal a biased distribution of the two alleles of these SNPs (Fig. 6).

Sequencing of the genomic DNA indicated that the two CD15s^bimodal individuals selected for the experiment were heterozygous for two coding SNPs (individual C: rs61147939 G/C and rs61739552 G/A; individual D: rs61147939 G/C and rs145035679 C/A) (Fig. 6a). Sequencing of PCR-amplified fragments from the *FUT6* cDNA of the basophils confirmed that each sub-fraction is dominated by either one of the two haplotypes: in the CD15s^high fraction of individual C, it is the G-G haplotype (rs61147939–rs61739552), while the CD15s^low fraction is dominated by the C-A haplotype. Likewise, for individual D it was G-C in the CD15s^high and C-A in the CD15s^low fraction (rs61147939–rs145035679). In a control experiment with *ST3GAL1*, which is not controlled by RME, equal amounts of the two alleles of a heterozygous SNP (rs1048479 G/A) were detected in the cDNA of the two basophil fractions (Fig. 6b). Thus, in basophils *FUT6* expression is indeed controlled by RME.

**Impact of *FUT6* null alleles on basophil counts and itch sensitivity.** The failure to roll on E-selectin-coated surfaces (Fig. 2 and Supplementary Movie 1) suggested that CD15s^low basophils

may have a compromised ability to extravasate through endothelial walls. As a consequence, their homing to sensitized sites in the periphery may be hindered and the cell may become "trapped" within the blood circulation. In order to determine whether the genetic defect actually translates into any of these functional deficiencies, we analyzed the data of two large genome-wide association studies (GWAS). Although in Europeans the tightness of rs778798-LB has slightly eroded ($r^2 \sim 0.8$) (Supplementary Figs. 5 and 6), a direct association of rs778798 with the basophil blood count in white blood cells had been reported[24]. The UK Biobank and INTERVAL study reporting these data was initially carried out with 173,480 participants. Genome-wide re-analysis of an increased database of 487,409 blood donors (Supplementary Fig. 8) confirmed the association with rs778798 ($p$ value = $8.86 \times 10^{-13}$) and furthermore revealed that also rs17855739, the tag SNP of the second linkage block, associates with the basophil counts in an independent fashion ($p$ value = $1.87 \times 10^{-03}$; Fig. 7a, left panel). For both SNPs, elevated basophil blood counts were observed for the minor alleles associated with the respective *FUT6* null allele. Their effect was additive as the p value for the association with the *FUT6* status (defined by both SNPs) further decreased to $2.39 \times 10^{-15}$. Correlation of the imputed FUT6 state with the mean adjusted basophil count revealed the lowest

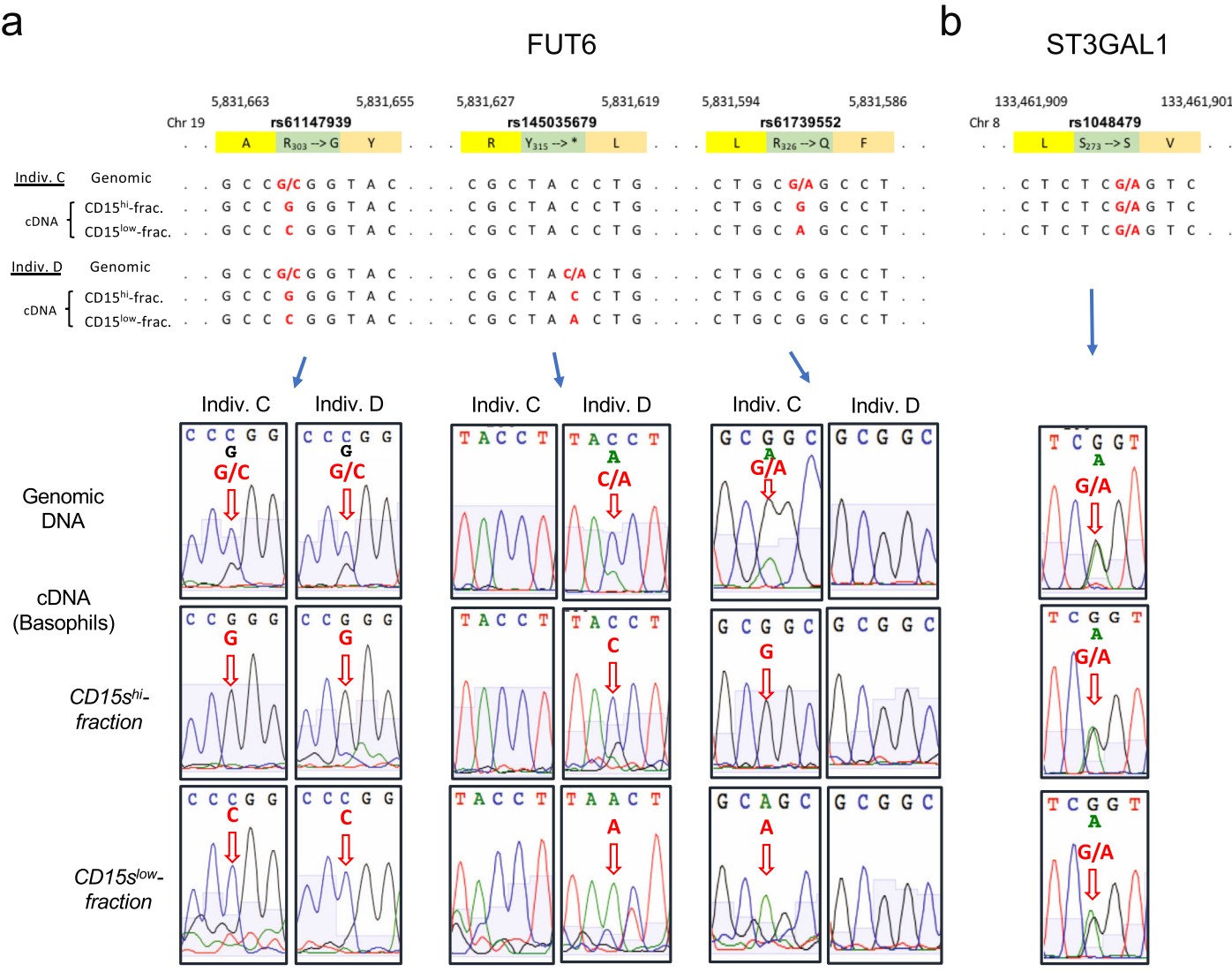

**Fig. 6 Random monoallelic expression of FUT6. a** FUT6 transcripts in basophils from the CD15s$^{high}$ and CD15s$^{low}$ fraction. cDNAs from the CD15s$^{high}$ and CD15s$^{low}$ basophil fraction of two CD15s$^{bimodal}$ individuals were isolated, and the region encoding for *FUT6* amplified by PCR and sequenced. The upper panel displays the *FUT6* sequence of the genomic DNA and the two cDNA regions covering three coding SNPs (rs61147939, rs145035679, and rs61739552). Each of the donors was heterozygous for two of these SNPs. Polymorphic bases are highlighted in red; base pair position as well as encoded amino acid residues are indicated. The histograms of the sequencing are displayed in the lower panels. Signals derived from polymorphic sites are indicated by red arrows. **b** *ST3GAL1* transcripts. As a control, cDNA sequencing of *ST3GAL1*, a gene not known to be controlled by random monoallelic expression, is shown. The donor was heterozygous for rs1048479. Only the sequence proximal to this SNP is shown.

number for *FUT6* +/+ (−0.0130), followed by *FUT6* −/+ (0.0085) and *FUT6* −/− (0.0264), confirming that the gene defect is indeed associated with increased basophil counts in the blood (Fig. 7a, right panel).

The impact on basophil-driven reactions in the periphery was tested with GWAS data on mosquito bites. Jones et al. had published a study in which 69,057 genotyped participants ranked their mosquito bite-induced itch sensitivity by a score of 0 (highest sensitivity) to 3 (lowest sensitivity)[25]. The study was carried out with data from 23andMe and revealed a direct association of rs778798. This was confirmed in a candidate approach, in which the SNP association was tested with an increased dataset of 614,354 participants (Supplementary Fig. 8). Besides the strong association with rs778798 ($p = 4.40 \times 10^{-63}$), the re-analysis also revealed the postulated association with rs17855739 ($p = 7.38 \times 10^{-12}$) (Fig. 7b, left panel). For both SNPs, the reduction in the itch score was associated with the respective minor allele and also here the association improved

when the test was carried out for the *FUT6* status defined by both SNPs ($p = 1.65 \times 10^{-79}$; Fig. 7b, left panel). The average itch score gradually increased from *FUT6* +/+ (2.28) over *FUT6* −/+ (2.30) to *FUT6* −/− (2.35) (Fig. 7b, right panel). Thus, individuals with defective *FUT6* genes have a reduced itch sensitivity for insect bites, a phenomenon presumably caused by a lower number of basophils reaching the peripheral sites.

**Impact of *FUT6* null alleles on allergy-related parameters.** As the main function of basophils is to mediate allergy and hypersensitivity reactions, *FUT6* deficiency should also correlate with allergy-related parameters. For our Singaporean SSIC cohort, we had collected already a large body of experimental, clinical, social, and demographic data[21,22]. Due to the high prevalence of house dust mite (HDM) allergies in Singapore, this included also a number of parameters related to respiratory allergies[21].

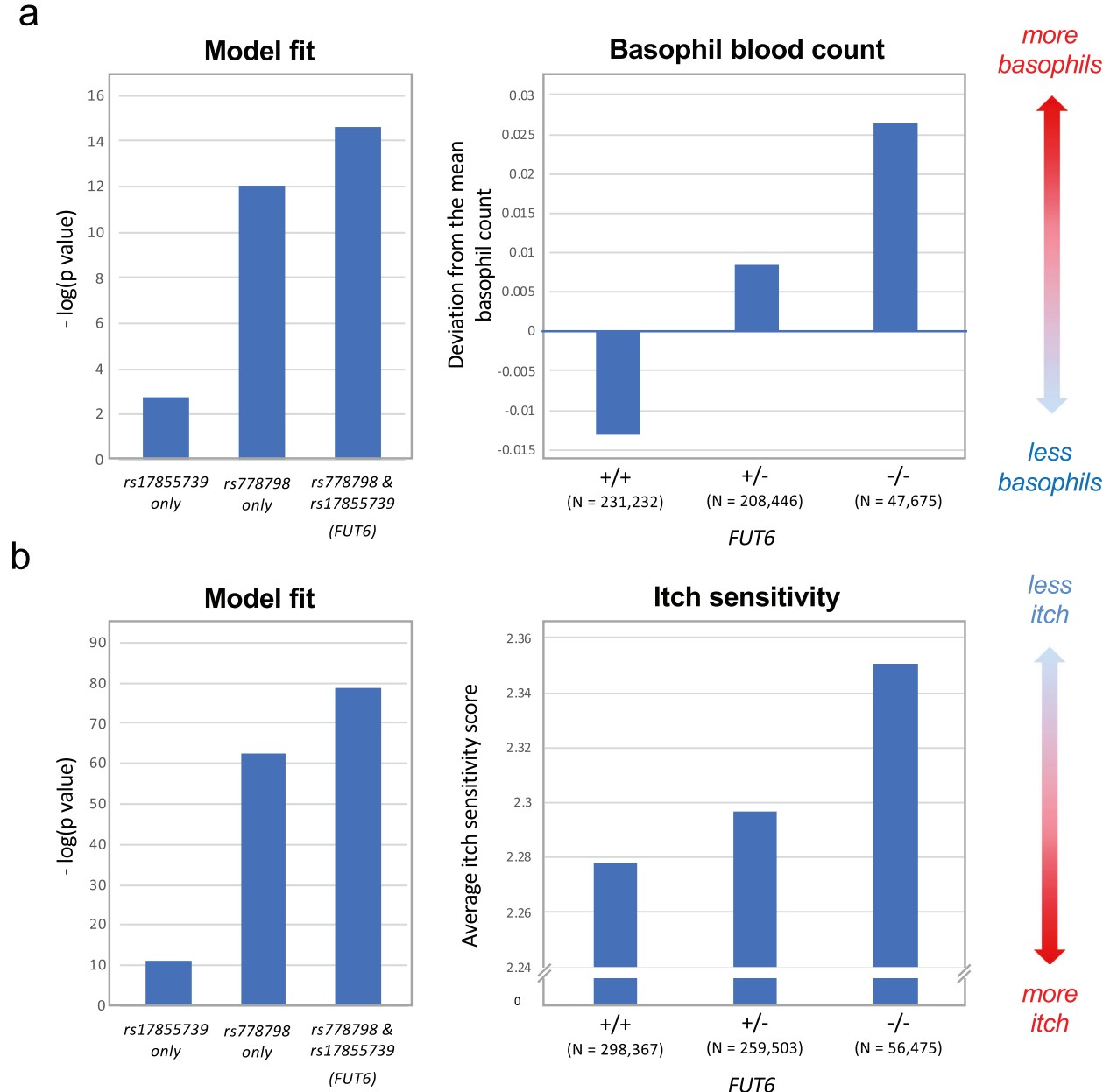

**Fig. 7 GWAS data on the association of *FUT6* null alleles to blood basophil counts and itch sensitivity. a** Basophil counts. The bar chart on the left shows the negative log10 *p* value of the association of the basophil blood count with rs17855739, rs778798, and the *FUT6* status imputed from these two SNPs (Fig. 4 and Supplementary Fig. 8). The bar chart on the right shows the genotype/phenotype correlation of the *FUT6* status (+/+, +/−, −/−) on the basophil blood count of 487,409 individuals of European ancestry, expressed as deviation from the mean basophil count. The data were generated by an extension of the study by Astle et al.[24]. **b** Mosquito bite-induced itch sensitivity. The bar chart on the left shows the negative log10 *p* value of the association of the itch sensitivity with rs17855739, rs778798, and the *FUT6* status defined by both SNPs (Supplementary Fig. 8). The box plot on the right shows the genotype/phenotype correlation of the implied *FUT6* status (+/+, +/−, −/−) on the average itch sensitivity score of 619,703 individuals of mostly European ancestry. The itch sensitivity score was defined by a questionnaire (0: maximum itch, 5: no itch). The data were provided by 23andMe and an extension of the study by Jones et al.[25].

When correlating this dataset with the state of *FUT6* defined by rs778798 and rs17855739, a nominal association was observed for allergy-related IgE parameters (Fig. 8a). This applied for allergen-specific IgE reacting against HDM ($p = 0.0055$, left panel) and to a lower extent also for total IgE ($p = 0.0402$, right panel). The median titer of HDM-specific IgE dropped from 16.75 kUA/L (*FUT6* +/+) to 8.13 kUA/L (*FUT6* +/−) to 0.84 kUA/L (*FUT6* −/−). The same trend was observed for the total IgE titer (Fig. 8b), which in Singapore largely mirrors the HDM-specific IgE titer[21]. In the same direction, a

nominal association was observed for eosinophil frequency (Fig. 8b). With a *p* value of 0.0186, the median frequency dropped from 1.08% (*FUT6* +/+) to 0.91% (*FUT6* +/−) to 0.62% (*FUT6* −/−). Notably, the same trend was also evident in the HDM-induced skin prick tests (Fig. 8c). A decrease in the response from *FUT6* +/+ over +/− to −/− was evident in both erythema ($p = 0.0126$, left panel) and wheal ($p = 0.0394$). Thus, the lack of sLe$^x$ expression on basophils caused by *FUT6* deficiency directly translates into functional defects of the IgE-mediated allergy response.

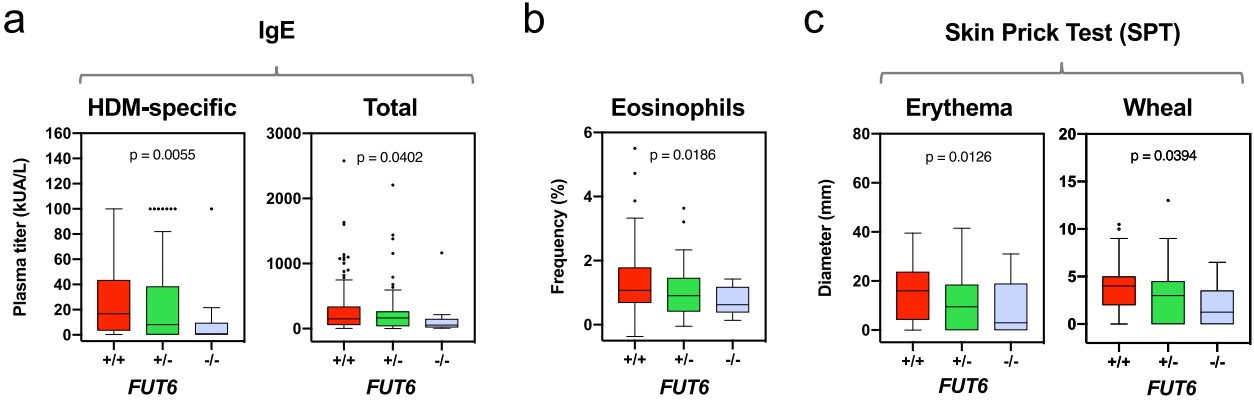

**Fig. 8 Association of *FUT6* null alleles to allergy-related parameters.** Plots were generated with data from individuals of the SSIC cohort[21,22]. **a** IgE. The box and whisker plots, box representing the upper and lower quartile, central line representing the median, and the minimum and maximum values indicate the distribution of the plasma amounts of allergen-specific IgE reactive to house dust mite (HDM) (left panel) and total IgE (right panel) in reference to the imputed *FUT6* status (+/+ (n = 124), +/− (n = 84), −/− (n = 14)). **b** Eosinophil frequency. The y-axis displays the frequency of eosinophils of the total leukocyte count. **c** Skin prick Test (SPT). The box and whisker plots, box representing the upper and lower quartile, central line representing the median, and the minimum and maximum values indicate the response of allergen-specific skin prick tests. The parameters displayed are the diameter of the erythema (left panel) and the wheal (right panel) induced by HDM. In all experiments, the significance was determined by Kruskal–Wallis test. The nominal p values are indicated in the figure.

## Discussion

FUT6 deficiency (OMIM #613852) was originally discovered as a congenital condition characterized by the lack of FUT6 activity in the plasma[7]. While the deficiency had been linked to *FUT6*, the functional and clinical significance remained unclear. In this study, we could show now that it selectively compromises function of basophils and pMCs. In individuals with defects in both *FUT6* alleles, the cells completely lack the expression of CD15s, a surface marker defined by the presence of sLe$^x$ glycan. Flow chamber experiments confirmed that lack of sLe$^x$ expression abrogates their ability to roll on E-selectin-coated surfaces. As a precondition for effective trans-endothelial migration, their extravasation appears to be compromised, which interferes with their ability to support allergic reactions. On the population level, this is evident in the increased blood basophil counts and a reduced itch sensitivity against insect bites. Lower levels of allergen-specific IgE, eosinophil counts, and allergen-induced skin prick test parameters further indicate that in particular the IgE-mediated response pathway is affected.

The striking cell type-specific manifestation of *FUT6* deficiency is caused by two independent factors. First, in contrast to neutrophils where the loss of sLe$^x$ expression can be compensated by the expression of orthologs such as VIM-2[4], the sLe$^x$ glycan in basophils cannot be replaced. Second, while most other leukocytes are using *FUT7* as a catalyst in the sLe$^x$ synthesis, basophils only employ *FUT6* in the synthesis pathway. It is thus the lack of other glycan paralogs combined with the unique use of FUT6 that is causing the cell-type specificity of the *FUT6* defect. The deficiency had been originally associated with two missense mutations, rs17855739 (E$_{247}$→K) and rs145035679 (Y$_{315}$→ stop)[7]. While it is still unclear whether any of the coding variants actually compromise the function of *FUT6*, our study at least confirmed that the derived alleles associate with some of the null alleles. Moreover, as we showed here, at least one other linkage block independently contributes to *FUT6* null alleles. The absence of any coding SNPs in rs778798-LB and a reduced mRNA levels detected in CD15s$^{low}$ cells of all tested donors suggest that besides the proposed coding defects transcriptional control strongly contributes to the *FUT6* gene defect.

Due to the high frequency of *FUT6* null alleles, between 5 and 20% of the population are *FUT6* −/−. The minor allele

frequencies (MAFs) of the two linkage blocks contributing to null alleles, however, varies strongly between the ethnic groups. The MAF of the linkage block tagged by the coding SNP rs17855739 is relatively low in Europeans (0.04) compared to East Asians (0.13), while the non-coding block tagged by rs778798 is especially high in European populations (East Asians: 0.09; Europeans: 0.27). Together the two linkage blocks account for a total of up to 50% defective *FUT6* genes (East Asians: 22%; Europeans: 31%; African: 36%; South Asian: 48%). While the non-coding linkage block tagged by rs778798 had already been associated with basophil count[24] and itch sensitivity[25], no GWAS associations have been reported for rs17855739.

Particularly striking is the bimodal expression pattern of the basophil CD15s expression. It is caused by RME, a rare and often cell type-specific phenomenon that affects <3% of the human genes[23]. In the case of FUT6, the stochastic inactivation of *FUT6* on one of the two sister chromatids results in the generation of mixed populations of sLe$^{x+}$ and sLe$^{x−}$ basophils in *FUT6* −/+ individuals. While the rolling capability of the sLe$^{x−}$ subset is clearly compromised, it is still unclear whether there are any other functional differences between the two basophil subsets. On the transcriptional level, we could not detect any differences beyond *FUT6* expression. However, sLe$^x$ is only one of the products generated by *FUT6*. Liu et al. have shown that the fucosylation of epidermal growth factor receptor by FUT6 dampens the ligand-induced activation by preventing receptor dimerization[26]. It is thus possible that in sLe$^{x−}$ basophils pathways other than migration are affected by the defect.

Basophils are an integral part of the hypersensitivity and allergic response. They also seem to play a nonredundant role in the acquired immunity against large ectoparasites, such as ticks[27], and exhibit effector functions by secreting histamine, leukotrienes, and the T helper type 2 (Th2) cytokines interleukin (IL)-4 and IL-13[28]. They also support the development of IgE-mediated inflammation[29] by boosting the Ig response[30]. IgE is often produced locally in the mucosa proximal to the site of sensitization[31]. Basophils may thus act as important regulators by releasing IgE-promoting Th2 cytokines in response to the allergen/IgE levels sensed by their FcεRI receptors[22].

The direct link between basophil function and IgE production was also evident in a prior study on basophil anergy[22]. Basophil

anergy is an acquired state observed in about 10% of the population, in which the signaling pathway of the entire basophil subset of the individual is blocked by the downregulation of their Syk kinase[32]. This inactivation was associated with a marked reduction in the levels of allergen-specific and total IgE, which was further linked with a reduced incidence of allergic airway disease[22]. A very similar outcome is observed when the functional integrity of basophils is compromised by *FUT6* deficiency. The homing defect, demonstrated in vitro in the failure to roll in flow chamber experiment and on the population level by the association of elevated basophil blood counts in carriers of *FUT6* null alleles, translated into lower titers of allergen-specific IgE as well as total IgE, a lower eosinophil count, and a reduced flare area after skin prick tests as well as a reduced itch sensitivity triggered by insect bites[25]. Both swelling and itching is caused by a cutaneous hypersensitivity reaction against mosquito salivary antigens. Another open question is what impact they have on mast cell function. Mature tissue-resident mast cells lack the expression of CD15s but the pMCs might require sLe$^x$ for proper homing to these sites.

In summary, *FUT6* selectively affects the function of basophils and pMCs by controlling the generation of the sLe$^x$ glycan (Supplementary Fig. 9). While we could show that *FUT6* deficiency has a dampening effect on the IgE-mediated allergic response, its high frequency in the general population indicates that the gene defect is apparently not associated with any other debilitating complications. Pharmaceutical interventions disrupting the sLe$^x$/E-selectin axis by blocking the sLe$^x$ synthesis through *FUT6* inhibition may thus be largely free of any side effects. As aberrant *FUT6* expression has also been associated with the metastasis of gastrointestinal cancer[33], it may thus represent a valuable new drug target not only for the treatment of allergic diseases but also for therapeutic interventions directed against tumors.

## Methods

**Cohort data**. This study was carried out with fresh blood samples obtained from healthy volunteers and frozen samples of PBMCs of the SSIC[21,22]. Of the 277 samples in the SSIC cohort, 229 samples of Chinese ethnicity were genotyped for this study. The cohort studies were done in accordance with the Helsinki declaration and approved by the Institutional Review Board at the National University of Singapore (IRB NUS 10-445) and the SingHealth Centralised Institutional Review Board (CIRB Ref: 2017/2806). Written informed consent was obtained from volunteer donors prior to sample collection.

**FACS analysis**. Samples of PBMCs that had been frozen in fetal bovine serum (FBS) containing 10% dimethyl sulfoxide were thawed in pre-warmed RPMI medium (Life Technologies), supplemented with 10% FBS using a Thawsome thawing device (Medax International, Inc., Salt Lake City, UT) as described[34]. To discriminate live from dead cells, PBMCs were incubated with the LIVE/DEAD Fixable Aqua Dead Kit (Life Technologies) in phosphate-buffered saline (PBS) for 10 min at room temperature. The cells were washed once with magnetic-activated cell sorting (MACS) buffer (0.5% bovine serum albumin (BSA), 2 mM EDTA in PBS) and stained for 15 min at 4 °C with the following cocktail of anti-human antibodies in MACS buffer: CD3 BV650 (UCHT1, BD Biosciences), CD14 FITC (61D3, eBioscience), CD15 APC (SSEA-1, Biolegend), CD15s BV421 (CSLEX1, BD Biosciences), CD16 eVolve 605 (EBIOCB16, eBioscience), CD19 BV786, (SJ25C1, BD Biosciences), CD25 PE-CF594 (M-A251, BD Biosciences), CD123 PerCP Cy5.5 (6H6; eBioscience), CD203 PC7 (97A6, Beckman Coulter), FcεRI PE (AER-37, eBioscience), and HLA-DR APC-H7 (L243; BD Biosciences). Cell pellets were washed once with MACS buffer, re-suspended in the 200 μL of the same buffer, and analyzed using the Fortessa flow cytometer (BD Biosciences). For staining of progenitor cells and pMCs, freshly isolated PBMCs were stained with CD3 FITC (UCHT1), CD14 FITC (61D3), CD16 FITC (eBioCB16, CB16), CD19 FITC (HIB19), CD56 FITC (MEM-188) lineage (LIN) markers, FcεRI BV605 (AER-37), CD117 APC (104D2), CD34 PE (581), and CD15s BV421 (CSLEX1). The gating strategy for progenitor cells and pMCs were based on previous study by Dahlin et al.[35]. Progenitor cells were identified as LIN− CD34+ and FcεRI−, and pMCs as LIN− CD34+ and FcεRI+.

**Basophil isolation**. Basophils for the rolling assay were isolated according to the manufacturer's instructions by negative selection using the Human Basophil Enrichment Kit (Stemcell). Briefly, approximately 40 ml of whole blood was collected in K$_3$EDTA vacutainer tubes (Greiner). Erythrocytes were first depleted using HetaSep (Stemcell) to obtain the nucleated cells, which were then incubated with the Human Basophil Enrichment Cocktail for 10 min at room temperature. The suspension was mixed with EasySep Nanoparticles at room temperature for 10 min. The cell suspension was added to a total volume of 10 ml and mixed by pipetting up and down 3–4 times. The tube containing the cell suspension was placed in the Silver EasySep Magnet and basophils were isolated by pouring. The number of basophils typically range from $3.6 \times 10^5$ to $4.5 \times 10^5$ and the purity of the basophil was determined to be 96.5–98.2%.

**Basophil rolling assay**. To prevent the Stemcell buffer (EDTA) in the basophil sample from interfering with the rolling assay, buffer exchange was first performed using dean flow fractionation (DFF)[36]. Sample and fresh sheath buffer (1× PBS with 0.1% BSA) were perfused through a 2-inlet, 4-outlet spiral microfluidic DFF device at an optimized flow rate of 110 μL/min and 1100 μL/min, respectively (ratio of 1:10), by separate syringe pumps (Chemyx Inc.). Basophil rolling was analyzed using a method previously described[36]. Briefly, 15 μL of DFF-purified basophils ($10^6$ cells/mL) was introduced to an E-selectin-coated (50 μg/mL, Peprotech) microchannel, and the sample was drawn in at a flow rate of 2.6 μL/min (~2 dyne/cm$^2$). Calcium chloride (20 μM, Sigma-Aldrich) was added to the sample to facilitate the calcium-dependent interaction. Phase-contrast images were captured at 0.5 s interval for 30 s at ×20 magnification using the MetaMorph software (Molecular Devices). Rolling speed of individual cell was measured using in-house MATLAB (Mathworks®) algorithm[36].

**Gene expression analysis**. The CD15s$^{high}$ and CD15s$^{low}$ basophil fractions of CD15$^{bimodal}$ individuals were isolated from PBMCs from 8 individuals by FACS sorting after staining with the following antibodies: CD14 PE-CF594 (MφP9, BD Biosciences), CD15s BV421 (CSLEX1, BD Biosciences), CD123 PerCP– Cy5.5 (6H6; eBioscience), FcεRI FITC (AER-37, eBioscience), and HLA-DR APC-H7 (L243; BD Biosciences). Cells were sorted using FACSAria II, FACSAria III, or Influx (BD Biosciences) and processed for RNA sequencing. Total RNA was isolated from CD15s-enriched cells using the ARCTURUS PicoPure RNA Isolation Kit (Applied Biosystems™ Thermo Fisher Scientific). Reverse transcription and amplification were performed using Smart-seq2 method[37]. Briefly, cDNA was synthesized from 2 ng of purified total RNA using modified oligo(dT) primers along with array control RNA spikes (Ambion® Thermo Fisher Scientific). To generate sufficient quantities of cDNA for downstream library preparation steps, 13 cycles of PCR amplification were performed. The quantity and integrity of cDNA was assessed using the DNA High Sensitivity Reagent Kit, Perkin Elmer LabChip GX (PerkinElmer, Waltham, MA, USA). Subsequently, pooled cDNA libraries were prepared (250 pg of cDNA per sample) using the Nextera XT kit (Illumina, San Diego, CA, USA) with dual indices for de-multiplexing. The libraries were quantified by qPCR (Kapa Biosystems) to ascertain the loading concentration. Samples were subjected to an indexed PE sequencing run of $2 \times 151$ cycles on an Illumina HiSeq 4000. The reads were mapped to the human genome GRCh38 using STAR and the gene counts were obtained using featureCounts (part of the Subread package). The gene counts were then analyzed for gene expression differences between CD15s$^{hi}$ vs. CD15s$^{lo}$ fractions from the same individual using a paired design in R version 3.6.3 using the edgeR package. Multiple testing correction was performed using the method of Benjamini and Hochberg. For the targeted analysis of *FUT6*, *FUT7*, and *GAPDH* gene expression, total RNA was first prepared from the isolated basophils using the TRIzol reagent (Thermo Fisher Scientific) and RNeasy Micro Kit (Qiagen) followed by cDNA synthesis using the QuantiTect Reverse Transcriptase Kit (Qiagen). Real-time qPCR was performed using SsoFast EvaGreen Supermix (Bio-Rad). The following primers were used for *FUT6* (Forward: 5'-ACAAACCCATAGCTCTGCC-3' and Reverse: 5'-TGAACCA-GATCCATCGCTGC-3'), *FUT7* (Forward 5'-ATCCTGGGAGACTGTGGATG-3' and Reverse: 5'-GTGCCAGACAAGGATGGTG-3'), and *GAPDH* (Forward: 5'-CAAGCTCATTTCCTGGTATGAC-3' and Reverse: 5'-GTGTGGTGGGGGACT-GAGTGTGG-3'). PCR conditions were as follows: enzyme activation (95°C, 30 s), followed by 40 cycles of denaturation (95°C, 5 s), annealing/extension (60°C, 10 s), and melt curve (65 min, 95°C, 10 s/step).

**RME analysis**. Allele-specific expression of *FUT6* was assessed by DNA sequencing of PCR fragments amplified from cDNAs of the CD15s$^{high}$ and CD15s$^{low}$ basophil fraction from CD15s$^{bimodal}$ donors together with matching samples of genomic DNA prepared from PBMCs. The analysis was carried out with primers that target a 300-bp region within the *FUT6* gene encompassing rs61147939, rs145035679, and rs61739552 (Forward: 5'-TTGCACCCCGACTACATCAC-3'; Reverse: 5'-GCGTGTCTGGTACCTCGGATT-3'). rs1048479 genotypes were determined with primers targeting the respective region in the *ST3GAL1* gene (Forward: 5'-GGAAGAGGCAGGCTAGGTCT-3'; Reverse: 5'-TCACTAAGGGC CGTGTCTTC-3'). All PCR products were purified using the Qiagen PCR Purification Kit and submitted for direct sequencing, using the respective forward and reverse primers to determine the sequence of both DNA strands.

**Genetic analysis and data correlation**. The linkage blocks rs778798-LB and rs17855739-LB were defined by the sets of perfectly linked SNPs detected in the

SSIC cohort. They were tagged by rs1531616 and rs17855739, respectively. The CD15s expression on basophils on biobanked PBMC samples of the SSIC cohort[21,22] was measured by FACS and associated with the genotype information using a linear model in MatrixEQTL[38]. Multiple testing correction was performed using the method of Benjamini and Hochberg. Linkage disequilibrium in the form of correlation coefficient was computed using PLINK 1.90b3.46. Genetic association of the genotype information to the SSIC parameters[21]HDM-specific IgE level, total IgE levels, eosinophil counts, and skin prick test results were done using PRISM 8.3.0. Additional linkage data for the Southern Han Chinese (CHS) and European population (EUR) was extracted from the "IDlink" webpage of the NIH (https://ldlink.nci.nih.gov).

**Genetic correlation of basophil counts**. Basophil count phenotype was collected and processed in UK Biobank as previously described[24]. Briefly, strict exclusion criteria were applied for people with relevant past and present disease, such as blood cancer. The phenotype was then adjusted by relevant covariates (sex, age, blood collections information, smoking status, alcohol consumption, etc.) and rank inverse normalized. The association of the *FUT6* SNPs rs1531616 and rs17855739 to the corrected basophil counts was tested using a linear regression model including the first ten principal components of a principal component analysis of the genetic data, in order to account for population structure. The following models were compared using analysis of variance: one including both SNPs, two models including each SNP separately, and the null model. The significance level was set to nominal significance ($p < 0.05$).

**Association with itch sensitivity**. The association of *FUT6* SNPs with mosquito-bite induced itch sensitivity was carried out by the personal genetics company 23andMe, Inc. using the imputed genotype data from 619,702 research participants of mostly European ancestry. The itch intensity was defined by responses to the question "When you are bitten by mosquitos, how much do the bites typically itch?" (0: "Very badly (impossible to ignore)"; 1: "Somewhat badly (definitely noticeable, at times hard to ignore)"; 2: "Only mildly (noticeable itching, but easy to ignore)"; 3: "Not at all (no noticeable itching)"). The itch intensity survey responses were scored in a numerical fashion (from 0 to 3) and analyzed using linear regression adjusting for age, sex, five principal components, and genotype platforms.

**Statistics and reproducibility**. Data were considered to be non-normally distributed and therefore non-parametric methods were applied for the statistical analysis unless otherwise stated. $p$ values <0.05 were considered to be statistically significant. The specifics of the statistical tests used are indicated in the figure legends.

**Reporting summary**. Further information on research design is available in the Nature Research Reporting Summary linked to this article.

## Data availability
Microarray data are deposited in National Center for Biotechnology Information (NCBI) Gene Expression Omnibus (GEO) and are accessible through GEO Series accession number GSE122281. Fitted linear models showing the contribution of rs778798 and rs17855739 to mosquito-induced itch sensitivity are provided in Supplementary Data 2. The linked genotype and phenotype data GWAS summary statistics are available from 23andMe at dataset-reqeust@23andMe.com but restrictions apply to the availability of these data, which were used under license for the current study, and so are not publicly available. The UK Biobank full summary statistics for the basophil counts GWAS are available here: ftp://ftp.sanger.ac.uk/pub/project/humgen/summary_statistics/UKBB_blood_cell_traits/baso.assoc. The raw genetic and phenotypic data from UK Biobank are available to all researchers upon application.

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

## Acknowledgements

This work was supported by grants from the Singapore Immunology Network (SIgN-06-006, SIgN-08-020, and SIgN-10-029), the National Medical Research Council (NMRC/1150/2008) Singapore, and Agency for Science, Technology and Research (A*STAR), Singapore. The SIgN Immunomonitoring platform supported by a BMRC IAF 311006 grant and BMRC transition funds #H16/99/b0/011. We specifically thank the blood donors of the Health Science Authority (HSA) of Singapore for providing leukocytes; 23andMe Research Participants, Ivy Low, Seri Munirah Mustafah, Nurhidaya Binte Shadan, and Ze Ming Lim from the SIgN Cytometry platform for help with the FACS sorting; Josephine Lum, Srinivasan KG, Shihui Foo, and Avery Khoo from SIgN Immunogenomics for the library preparation and sequencing of the RNA samples; Rene Hennig from the MPI in Magdeburg for discussions and critical reading of the manuscript; and members of the 23andMe Research Team: Michelle Agee, Stella Aslibekyan, Adam Auton, Robert K. Bell, Katarzyna Bryc, Sarah K. Clark, Sarah L. Elson, Kipper Fletez-Brant, Pierre Fontanillas, Nicholas A. Furlotte, Pooja M. Gandhi, Karl Heilbron, Barry Hicks, Karen E. Huber, Ethan M. Jewett, Yunxuan Jiang, Aaron Kleinman, Keng-Han Lin, Nadia K. Litterman, Matthew H. McIntyre, Kimberly F. McManus, Joanna L. Mountain, Sahar V. Mozaffari, Priyanka Nandakumar, Elizabeth S. Noblin, Carrie A.M. Northover, Jared O'Connell, Steven J. Pitts, G. David Poznik, J. Fah Sathirapongsasuti, Janie F. Shelton, Suyash Shringarpure, Joyce Y. Tung, Robert J. Tunney, Vladimir Vacic, and Xin Wang.

## Author contributions

K.J.P., B.S.L., N.Y., D.K., W.L., J.D.C., and T.D. performed the experiments on phenotyping and functional and genetic characterization. H.W.H. carried out the flow chamber experiments. A.K.A. and D.Y.W. organized the collection and analysis of cohort samples K.J.P., B.L., M.P., and O.R. analyzed and interpreted the data. S.C. and E.R. carried out carbohydrate analysis. D.V. and N.S. tested replication of the GWAS on the basophil counts and C.T., Y.J., and the 23andMe Research Team tested replication of the GWAS on the itch sensitivity against mosquito bites. O.R. supervised the project and wrote the manuscript.

## Competing interests

C.T., Y.J., and members of the 23andMe Research Team are employees of 23andMe, Inc. and hold stock or stock options in 23andMe. The remaining authors have no competing interests.

## Additional information

## the 23andMe Research Team

Michelle Agee[6], Stella Aslibekyan[6], Adam Auton[6], Elizabeth Babalola[6], Robert K. Bell[6], Jessica Bielenberg[6], Katarzyna Bryc[6], Emily Bullis[6], Briana Cameron[6], Daniella Coker[6], Gabriel Cuellar Partida[6], Devika Dhamija[6], Sayantan Das[6], Sarah L. Elson[6], Teresa Filshtein[6], Kipper Fletez-Brant[6], Pierre Fontanillas[6], Will Freyman[6], Pooja M. Gandhi[6], Karl Heilbron[6], Barry Hicks[6], David A. Hinds[6], Karen E. Huber[6], Ethan M. Jewett[6], , Yunxuan Jiang[6], Aaron Kleinman[6], Katelyn Kukar[6], Vanessa Lane[6], Keng-Han Lin[6], Maya Lowe[6], Marie K. Luff[6], Jennifer C. McCreight[6], Matthew H. McIntyre[6], Kimberly F. McManus[6], Steven J. Micheletti[6], Meghan E. Moreno[6], Joanna L. Mountain[6], Sahar V. Mozaffari[6], Priyanka Nandakumar[6], Elizabeth S. Noblin[6], Jared O'Connell[6], Aaron A. Petrakovitz[6], G. David Poznik[6], Morgan Schumacher[6], Anjali J. Shastri[6], Janie F. Shelton[6], Jingchunzi Shi[6], Suyash Shringarpure[6], & Chao Tian[6], Vinh Tran[6], Joyce Y. Tung[6], Xin Wang[6], Wei Wang[6], Catherine H. Weldon[6] & Peter Wilton[6]

