## [Peer Review File · Communications Biology]

Reviewers' comments:

Reviewer #1 (Remarks to the Author):

In this study the authors present evidence that null variants in FUT6 correlate (and presumably cause) low sialyl-Lewis x expression selectively on basophils, which cannot compensate as other cell types do. They also present evidence that this selectively compromises basophil function and is associated with allergic responses.

My comments are primarily on the human genetics aspects of the manuscript:

Population stratification. From the methods section, it appears that the authors do not control for population stratification in the association studies of SSIC. This raises the possibility of artifactual associations due to ancestry. I suggest they include genotype PCs, as is standard in the field (see EIGENSTRAT, flashPCA, etc). This is of particular importance in Singaporean populations, which have three distinct ancestral backgrounds.

The authors claim that rs778798 and rs17855739 have independent effects on basophil-specific CD15s expression (line 340), because they are in low LD. Whilst this is likely true, the more usual way to demonstrate this in association studies is through conditional analysis, where the most strongly associated SNP is included as a covariate in a linear regression association model. This is already implemented in plink, which the authors are using for their work.

Absence of minor allele combinations of rs778798 and rs17855739 (line 351). The authors conclude that the minor allele homozygote combinations must not exist because they do not find them. However, from 1,000 Genomes data, both variants have a minor allele frequency (MAF) around 0.14 in East Asian populations. The expected frequency for each minor homozygote is therefore $0.14 * 0.14 \sim 0.02$. As the two SNPs are effectively in linkage equilibrium ($r^2 < 0.5$ in their sample), the expected frequency of the AA/TT haplotype would be $0.02 * 0.02 = 0.0004$, or 1/2500. The absence of this haplotype, therefore, cannot be inferred to be due to functional selection but to low frequency.

IgE correlation (line 450 onwards). The methods used here are unclear to me (they do not appear covered in the methods section). Are the reported p values corrected for multiple testing? How many phenotypes in the same cohort did the authors look at to arrive at these associations? Is this analysis corrected for population stratification?

The authors are in effect conducting an epistasis test (3x3 chi square) - they may wish to acknowledge this.

There are standard ways to display linkage disequilibrium between markers (Figures 4B and S4) - usually as lower triangles shaded by r^2 . The current display is somewhat confusing.

FUT6 does not appear to be under lineage constraint as detectable in gnomAD.

Polymorphisms present in the general population are generally referred to as variants, rather than mutations (line 495, 496 etc)

Reviewer #2 (Remarks to the Author):

This manuscript by Puan et al. demonstrates the critical effect of FUT6 on the generation of E-selectin ligands on basophils and, consequently, on the rolling interaction of these cells with endothelium and their extravasation. Beginning with the observation of variable basophil CD15s staining between individuals, this study goes on to precisely define the gene and SNPs responsible for this effect, demonstrates random mono-allelic expression of FUT6, and correlates these FUT6 null alleles with itch sensitivity scores and allergy-related measures. While one of the SNPs had previously been associated with basophil counts, the genetic control of basophil extravasation and

the significance of this ability in the regulation of type 2 immunity in the general human population is novel and potentially important.

The comments that I have for the authors are listed below:

1. The conclusion that the effect is limited to basophils is not established in a compelling manner. The investigation of other leukocytes appears to have been somewhat superficial, focusing only on three cell types other than basophils. Indeed, there may be a small population of cells within those labeled "others" that likewise lost CD15s staining in Fig. 1B. Notably, although the manuscript correctly remarks that CD15s is absent on mature mast cells, mast cell progenitors express E-selectin ligands (Boyce, Mellor et al. 2002). A more thorough investigation of other cell types and whether they are directly affected by the loss of FUT6 is warranted or the language surrounding the cell specificity of FUT6 deficiency should be softened.
2. On line 490, a reference is made to "VIM1" expression on neutrophils. I believe this should be "VIM-2," as noted earlier in the manuscript.
3. The legend for Figure 1 does not match the figure. The descriptions of Fig. 1 B and C appear to be switched.
4. The y-axis label for the right panel of Fig. 7A should more precisely reflect the numeric values, e.g., "Deviation from the mean basophil count."

Reference:

Boyce, J. A., E. A. Mellor, B. Perkins, Y. C. Lim and F. W. Luscinskas (2002). "Human mast cell progenitors use alpha4-integrin, VCAM-1, and PSGL-1 E-selectin for adhesive interactions with human vascular endothelium under flow conditions." *Blood* 99(8): 2890-2896.

Reviewer #3 (Remarks to the Author):

Overall this is a very interesting report showing that deficiency in a human fucosyltransferase, FUT6, results in a unique basophil phenotype resulting from a loss of the glycan epitope sialyl-Lewis X (sLex). This sLex epitope (NeuAc α 2-3Gal β 1-4(Fuc α 1-3)GlcNAc) is found as a terminal sequence on glycans of glycoproteins on the surface of many white blood cells and is the ligand recognized by cell surface receptors called selectins that are expressed on inflamed endothelial cells and mediate trafficking of cells from the blood into inflamed tissues. The authors convincingly show that humans with mutations that result in FUT6 deficiency have little or no sLex expressed on the surface of their basophils, and as a result have higher basophil levels in their blood, presumably because they can't 'get out' into tissues through a selectin mediated process. Clinically, this results in reduced basophil responses, as particularly documented by reduced itch to insect bites.

The human genetics aspects are well done, and the identification of FUT6 mutations that result in reduced sLex expression in basophils are compelling. Of significant interest is that individuals who are heterozygous for the null mutations often have bimodal expression of sLex, which is accounted for by random allelic exclusion so that some basophils express the FUT6 gene and have normal levels of sLex, and other neutrophils express no FUT6 and have little or no sLex.

The authors note that a defective FUT6 gene results in sLex deficiency in basophils but not other leukocyte cell types (e.g. neutrophils). The major concern of this referee is a superficial analysis of the role of FUT6 in the biosynthesis of sLex in basophils, and the clear differences between the role of FUT6 in production of sLex in basophils and other white blood cells.

Major points:

1. Line 67. The authors note that there are six FUTs that can produce the Fuc α 1-3GlcNAc linkage found in sLex, and that only FUT6 and FUT7 are involved in production of E-selectin ligands (e.g. sLex). However, the reference cited only talks about FUT7, and clearly states that FUT7 is the dominant gene in production of sLex in leukocytes. Are there other reports that state that FUT6 can produce sLex in leukocytes? If so, they should be cited.

2. Line 71. It is stated that deficiencies in FUT6 and FUT7 are without any phenotypes, but there are no citations to support this statement.

3. Line 305-307 and 311-312 and Figure 3. It is stated that FUT7 is expressed at equally high levels in both sLex⁺ and sLex⁻ basophils. If FUT7 is expressed in basophils, and it is arguably the dominant enzyme for synthesis of sLex, why isn't sLex expressed in basophils? This is neither explained or raised as evidence contradictory to their hypothesis.

4. Line 305-307 and 311-312, Figure 3 and Figure S1. It is not clear why ST3Gal1 is included as being relevant to the synthesis of sLex. It does not transfer sialic acid to the structure shown in Fig. S1. It transfers sialic acid exclusively to the O-linked glycan Gal β 1-3GalNAc α Thr/Ser. It does not transfer sialic acid to Gal β 1-4GlcNAc-R needed to make sLex. So the statements regarding ST3Gal1 are in error.

Minor points:

5. Line 82. Should Javenese be Japanese?

6. Figure 5. The data are very nice, but the legend is very unclear. The first sentence is : FUT6 states. What does that mean? Should be a more descriptive legend title.

7. Figure 5. In panel B y-axis, what does "Absolute count" mean. Based on the data in panel A, it most likely means the number of individuals that were analyzed. Should be clearly explained in the legend.

Reviewers' comments:

Reviewer #1 (Remarks to the Author):

In this study the authors present evidence that null variants in FUT6 correlate (and presumably cause) low sialyl-Lewis x expression selectively on basophils, which cannot compensate as other cell types do. They also present evidence that this selectively compromises basophil function and is associated with allergic responses.

My comments are primarily on the human genetics aspects of the manuscript:

1.1. Population stratification. From the methods section, it appears that the authors do not control for population stratification in the association studies of SSIC. This raises the possibility of artifactual associations due to ancestry. I suggest they include genotype PCs, as is standard in the field (see EIGENSTRAT, flashPCA, etc). This is of particular importance in Singaporean populations, which have three distinct ancestral backgrounds.

While it is certainly true that Singapore has three major ethnicities, our Singapore cohort consists exclusively of individuals of Chinese ethnicity. It is now explicitly stated in Line 114-115 in Materials and Methods as well as in Line 808-809 in the legend to Figure 4.

1.2. The authors claim that rs778798 and rs17855739 have independent effects on basophil-specific CD15s expression (line 340), because they are in low LD. Whilst this is likely true, the more usual way to demonstrate this in association studies is through conditional analysis, where the most strongly associated SNP is included as a covariate in a linear regression association model. This is already implemented in plink, which the authors are using for their work.

We thank the referee for this suggestion. To show that rs778798 and rs178555739 have independent effects on basophil specific CD15s expression, we included the conditional analysis using the respective plink tool.

The relevant non-conditional results are as follows:

CHR	SNP	BP	A1	TEST	NMISS	BETA	STAT	P
19	chr19:5839613:A:C	5839613	A	ADD	219	-1.046	-8.606	1.54E-15
19	chr19:5831840:C:T	5831840	T	ADD	220	-0.9642	-7.061	2.19E-11

The conditional results conditional on rs778798 are as follows:

CHR	SNP	BP	A1	TEST	NMISS	BETA	STAT	P
19	chr19:5831840:C:T	5831840	T	ADD	215	-1.089	-10.21	3.73E-20
19	chr19:5831840:C:T	5831840	T	chr19:5839613:A:C	215	-1.153	-11.4	8.82E-24

The rs17855739 is still significant after conditional for rs778798.

The conditional results conditional on rs17855739 are as follows:

CHR	SNP	BP	A1	TEST	NMISS	BETA	STAT	P
19	chr19:5839613:A:C	5839613	A	ADD	215	-1.153	-11.4	8.82E-24
19	chr19:5839613:A:C	5839613	A	chr19:5831840:C:T	215	-1.089	-10.21	3.73E-20

The rs778798 is still significant after conditional for rs17855739.

The conditional analysis shows rs778798 and rs17855739 have indeed independent effects on basophil-specific CD15s expression. We added these tables in the Supplementary Info (new Supplementary Figure 7a) and also mention it now in the main text:

Line 365-368. “The r^2 value between rs778798 and rs17855739 was <0.05 , suggesting that each of the two linkage blocks had an independent effect on the basophil-specific CD15s expression. This assumption was indeed confirmed after carrying out a conditional SNP analysis (new Supplementary Figure 7a).”

1.3. Absence of minor allele combinations of rs778798 and rs17855739 (line 351). The authors conclude that the minor allele homozygote combinations must not exist because they do not find them. However, from 1,000 Genomes data, both variants have a minor allele frequency (MAF) around 0.14 in East Asian populations. The expected frequency for each minor homozygote is therefore $0.14 * 0.14 \sim 0.02$. As the two SNPs are effectively in linkage equilibrium ($r^2 < 0.5$ in their sample), the expected frequency of the AA/TT haplotype would be $0.02 * 0.02 = 0.0004$, or 1/2500. The absence of this haplotype, therefore, cannot be inferred to be due to functional selection but to low frequency.

We agree with the reviewer that the absence of the haplotype containing a combination of the two minor allele is not indicative of a functional selection. We don't think though that the two SNPs are in linkage equilibrium. While r^2 is indeed < 0.05 , D' equals 1. This applies not only for the small Singapore cohort ($n=229$; $D' = 1.00$) but also for the two larger (but ethnically more diverse) cohorts from Sanger ($n=487,409$; $D' = 0.99$) and 23andMe ($n=614,354$; $D' = 0.99$). Given the size of these cohorts of about 500,000 individuals we would expect at least 200 carriers with the AA/TT genotype, while none was detected in any of the two cohorts. We included the r^2 and D' values in the new Supplementary Figure 7b and added the following sentence:

Line 390-391: “This same trend is also evident in two larger but ethnically more diverse cohorts from the UK Biobank and INTERVAL study and 23andMe. While both cohorts comprised around 500,000 individuals, AA/TT was completely absent, which is reflected in D' values of 0.99 for both cohorts (Supplementary Fig. 7b). Thus, a haplotype combining the derived alleles of both linkage blocks on one strand virtually does not exist.”

1.4. IgE correlation (line 450 onwards). The methods used here are unclear to me (they do not appear covered in the methods section). Are the reported p values corrected for multiple testing? How many phenotypes in the same cohort did the authors look at to arrive at these associations? Is this analysis corrected for population stratification?

Allergic parameters including total IgE, specific-IgE, eosinophils and wheal reaction size were performed using Kruskal-Wallis test. The p values have not been corrected for multiple testing. To clarify this we stated now in the legend to Figure 8 that these are nominal p value line 863.

1.5. The authors are in effect conducting an epistasis test (3x3 chi square) - they may wish to acknowledge this.

While we are showing the 3x3 matrix we are actually not carrying out a chi square analysis. Epistasis implies that an apparent linkage disequilibrium is due to the functional interaction of SNPs but it is caused here by the absence of the A-T haplotype. The figure is just meant to illustrate this fact. To strengthen our case, we included the 3x3 matrix for the two larger cohort in the new Supplementary Figure 7b.

1.6. There are standard ways to display linkage disequilibrium between markers (Figures 4B and S4) - usually as lower triangles shaded by r2. The current display is somewhat confusing.

The triangular display of the linkage disequilibrium between the SNPs of the two FUT6 haplotypes are now depicted in the new Supplementary Figure 5.

1.7. FUT6 does not appear to be under lineage constraint as detectable in gnomAD.

We have checked FUT6 in the gnomAD browser. As there is indeed no difference in the Expected variant counts and Observed variant counts, it confirms that the FUT6 polymorphism was not promoted by selection pressure. We acknowledged this by adding the following sentence:

Line 384: "FUT6 does not appear to be under lineage constraint as detectable in gnomAD (<https://gnomad.broadinstitute.org/>) and for both linkage blocks the ancestral allele is represented by the major allele."

1.8. Polymorphisms present in the general population are generally referred to as variants, rather than mutations (line 495, 496 etc)

We have changed 'deleterious mutations' to 'deleterious variants' in line 387, 'coding mutations' to 'coding variants' in line 533, and 'FUT6 mutations' to 'FUT6 variants' the figure legend in new Supplementary Figure 9.

Reviewer #2 (Remarks to the Author):

This manuscript by Puan et al. demonstrates the critical effect of FUT6 on the generation of E-

selectin ligands on basophils and, consequently, on the rolling interaction of these cells with endothelium and their extravasation. Beginning with the observation of variable basophil CD15s staining between individuals, this study goes on to precisely define the gene and SNPs responsible for this effect, demonstrates random mono-allelic expression of FUT6, and correlates these FUT6 null alleles with itch sensitivity scores and allergy-related measures. While one of the SNPs had previously been associated with basophil counts, the genetic control of basophil extravasation and the significance of this ability in the regulation of type 2 immunity in the general human population is novel and potentially important.

The comments that I have for the authors are listed below:

2.1. The conclusion that the effect is limited to basophils is not established in a compelling manner. The investigation of other leukocytes appears to have been somewhat superficial, focusing only on three cell types other than basophils. Indeed, there may be a small population of cells within those labeled “others” that likewise lost CD15s staining in Fig. 1B.

In support of our claim that that the CD15s phenotype is essentially restricted to basophils we included now RNA sequencing data of 30 Leukocyte subsets in the new Supplementary Figure 2. The figure was generated based on the published data from Monaco G. et al. (Cell Rep. 2019 26(6): 1627–1640). It confirms the high expression of FUT6 in basophils while, with the exception of progenitor cells, it is undetectable in virtually all other cells tested. We acknowledged that in the main text by adding:

Line 331-336: “A very similar result was obtained when analyzing the published RNA-sequencing data from Monaco et al.²⁶ (Supplementary Fig. 2). Also, in this study the amount of FUT6 and FUT7 RNA in basophils was roughly comparable. More importantly, while FUT7 was expressed in every cell analyzed, FUT6 was detected at substantial amounts only in basophils and progenitor cells, a finding in-line with the suggested cell type-specific of the phenotype variations of CD15s.”

2.2. Notably, although the manuscript correctly remarks that CD15s is absent on mature mast cells, mast cell progenitors express E-selectin ligands (Boyce, Mellor et al. 2002). A more thorough investigation of other cell types and whether they are directly affected by the loss of FUT6 is warranted or the language surrounding the cell specificity of FUT6 deficiency should be softened.

We thank the referee for suggesting to look at mast cell progenitors (pMC). This population is extremely rare (0.002 – 0.004 % of the PBMC) and characterized by the expression of CD34, CD117 and FCERI. The analysis of a set of newly added FACS experiments revealed that pMC show the same CD15s phenotype variations as the basophils. We included the data in a new Supplementary Figure 3 and added the following line to the main text and included the reference by Boyce et al. (ref. 27):

Line 336-334: “When analyzing the CD34+ CD117+ progenitor pool of the PBMC compartment by FACS, we noted that only FCER1+ mast cell progenitors (pMC)²⁷ exhibit the same CD15s variation as basophils, pMC are a tiny population comprising only approximately mean 0.003% of the PBMC (range 0.002% - 0.004%) and the phenotype was not shared by the bulk of FCERI- progenitors (Supplementary Fig. 3)”

We also acknowledged this in line 38-39 of the abstract, line 96-97 of the introduction and line 585-587 and line 588-589 in the discussion. The following section is also added to the discussion:

Line 585 “Another open question is what impact they have on mast cell function. Mature tissue-resident mast cells lack the expression of CD15s but the mast cell progenitors might require sLe^x for proper homing to these sites.”

2.3. On line 490, a reference is made to “VIM1” expression on neutrophils. I believe this should be “VIM-2,” as noted earlier in the manuscript.

This is correct. The incorrect “VIM1” has been changed to “VIM-2”.

2.4. The legend for Figure 1 does not match the figure. The descriptions of Fig. 1 B and C appear to be switched.

Thanks for pointing this out. The descriptions of Fig. 1b and b have been corrected in the revised manuscript.

2.5. The y-axis label for the right panel of Fig. 7A should more precisely reflect the numeric values, e.g., “Deviation from the mean basophil count.”

We have corrected y-axis label in the revised manuscript.

Reference:

Boyce, J. A., E. A. Mellor, B. Perkins, Y. C. Lim and F. W. Luscinskas (2002). "Human mast cell progenitors use alpha4-integrin, VCAM-1, and PSGL-1 E-selectin for adhesive interactions with human vascular endothelium under flow conditions." *Blood* 99(8): 2890-2896.

Reviewer #3 (Remarks to the Author):

Overall this is a very interesting report showing that deficiency in a human fucosyltransferase, FUT6, results in a unique basophil phenotype resulting from a loss of the glycan epitope sialyl-Lewis X (sLex). This sLex epitope (NeuAc α 2-3Gal β 1-4(Fuc α 1-3)GlcNAc) is found as a terminal sequence on glycans of glycoproteins on the surface of many white blood cells and is the ligand recognized by cell surface receptors called selectins that are expressed on inflamed endothelial cells and mediate trafficking of cells from the blood into inflamed tissues. The authors convincingly show that humans with mutations that result in FUT6 deficiency have little or no sLex expressed on the surface of their basophils, and as a result have higher basophil levels in their blood, presumably because they can't 'get out' into tissues through a selectin mediated process. Clinically, this results in reduced basophil responses, as particularly documented by reduced itch to insect bites.

The human genetics aspects are well done, and the identification of FUT6 mutations that result in reduced sLex expression in basophils are compelling. Of significant interest is that individuals who are heterozygous for the null mutations often have bimodal expression of sLex, which is accounted for by random allelic exclusion so that some basophils express the FUT6 gene and have normal levels of sLex, and other neutrophils express no FUT6 and have little or no sLex.

The authors note that a defective FUT6 gene results in sLex deficiency in basophils but not other leukocyte cell types (e.g. neutrophils). The major concern of this referee is a superficial analysis of the role of FUT6 in the biosynthesis of sLex in basophils, and the clear differences between the role of FUT6 in production of sLex in basophils and other white blood cells.

Major points:

3.1. Line 67. The authors note that there are six FUTs that can produce the Fuca1-3GlcNAc linkage found in sLex, and that only FUT6 and FUT7 are involved in production of E-selectin ligands (e.g. sLex). However, the reference cited only talks about FUT7, and clearly states that FUT7 is the dominant gene in production of sLex in leukocytes. Are there other reports that state that FUT6 can produce sLex in leukocytes? If so, they should be cited.

Thanks for pointing this out. As you may have noticed, glycan biology is not our prime field of expertise, so I want to apologize for some of the accidental errors that were made in writing of the manuscript. While both FUT6 and FUT7 are involved the generation of sLe^x, the referee is right that FUT7 is the dominant enzyme for leukocytes. FUT6 was known to be involved in the sLe^x production on plasma proteins and epithelial cancer cells. To clarify this, we now added the following sentence to the introduction and added with refs. 6 (Kudo et al.) and 8 (Kannagi et al.) two new references:

Line 70: "FUT7 is reportedly the dominant transferase for leukocytes^{1,6}, whereas FUT6 is only known to be involved in the sLe^x production on plasma proteins and epithelial cancer cells^{7,8}"

3.2. Line 71. It is stated that deficiencies in FUT6 and FUT7 are without any phenotypes, but there are no citations to support this statement.

To clarify the statement, we changed "phenotypes" in line 74 with "apparent clinical consequences" and quoted references 7 and 10 as evidence.

3.3. Line 305-307 and 311-312 and Figure 3. It is stated that FUT7 is expressed at equally high levels in both sLex+ and sLex- basophils. If FUT7 is expressed in basophils, and it is arguably the dominant enzyme for synthesis of sLex, why isn't sLex expressed in basophils? This is neither explained or raised as evidence contradictory to their hypothesis.

This was indeed puzzling for us, too. However, the RNA-seq data from Monaco et al. (Cell Rep. 2019 26(6): 1627–1640) now adds some clarity. According to this Neutrophils, whose sLe^x expression depends solely on FUT7, lack expression of FUT6 but express about 30 times higher amounts of FUT7 mRNA compared to basophils. We added this data now in then new **Supplementary Figure 3** and added the following sentence:

Line 342-344: "Notably, while neutrophils lack any detectable expression of FUT6 RNA, their levels of FUT7 were more than 30-fold higher compared to basophils, explaining the tight link between CD15s and FUT7 for these cells (**Supplementary Fig. 2**)."

3.4. Line 305-307 and 311-312, Figure 3 and Figure S1. It is not clear why ST3Gal1 is included as being relevant to the synthesis of sLex. It does not transfer sialic acid to the structure shown in Fig. S1. It transfers sialic acid exclusively to the O-linked glycan Galβ1-3GalNAcαThr/Ser. It does not transfer sialic acid to Galβ1-4GlcNAc-R needed to make sLex. So the statements regarding ST3Gal1 are in error.

This was indeed a lapse on our side (which thankfully is without major consequences). As ST3Gal1 served only as a redundant control, we removed it from Figure 3 and delete all sentences relating to that gene from both the Figure Legend and the main text (line 785, line 321-322 and line 327). We

kept it though for Figure 6b as it served here only as an example of a polymorphism not affected by RME.

Minor points:

3.5. Line 82. Should Javenese be Japanese?

The original statement was correct as the study was really done with a small cohort from Java, Indonesia.

3.6. Figure 5. The data are very nice, but the legend is very unclear. The first sentence is : FUT6 states. What does that mean? Should be a more descriptive legend title.

We have revised the title of the figure legend in line 811 to “Association of FUT6 null alleles with the bimodal sLe^x expression in basophils”

3.7. Figure 5. In panel B y-axis, what does “Absolute count” mean. Based on the data in panel A, it most likely means the number of individuals that were analyzed. Should be clearly explained in the legend.

The y-axis depicts the number of individuals in SSIC cohort. We have revised the y-axis title to “Number of individuals.”

REVIEWERS' COMMENTS:

Reviewer #1 (Remarks to the Author):

The authors have done a substantial amount of work in addressing my comments. My only remaining comment is about the discussion of the data in Figure 8 starting on line 500. Given that the p-values shown are nominal and most would not survive correction for the multiple phenotypes tested, I suggest the authors moderate the strengths of their claims of significance.

Reviewer #2 (Remarks to the Author):

The manuscript by Puan et al. demonstrates the critical effect of FUT6 on the generation of E-selectin ligands on basophils and, in the revised version, on mast cell progenitors. SNPs in the gene are demonstrated to affect FUT6 levels and the rolling interaction of basophils with endothelium in a general population cohort. Furthermore, due to random mono-allelic expression of FUT6, distinct populations of CD15s+ and CD15s- basophils and mast cell progenitors can be observed in individuals with only a single FUT6 null allele. The correlation of FUT6 null alleles with allergy- and itch-related measures and the likely connection with basophil and mast cell progenitor extravasation are important and novel contributions to the field. The authors have addressed my concerns with the original manuscript, and I have no further criticisms to offer.

Reviewer #3 (Remarks to the Author):

The authors have considered and addressed all the concerns raised by this reviewer.

REVIEWERS' COMMENTS:

Reviewer #1 (Remarks to the Author):

The authors have done a substantial amount of work in addressing my comments. My only remaining comment is about the discussion of the data in Figure 8 starting on line 500. Given that the p-values shown are nominal and most would not survive correction for the multiple phenotypes tested, I suggest the authors moderate the strengths of their claims of significance.

We noted reviewer comments and as such we have rephrased our claims:

line 496, "..., from a strong correlation was observed for allergy-related Ig-E (Fig. 8a)." to "..., a nominal association was observed for allergy-related Ig-E parameters (Fig. 8a)."

Line 499 from "... a significant correlation was observed for eosinophil frequency (Fig. 8b)." to "... , a nominal association was observed for eosinophil frequency (Fig. 8b)."

Reviewer #2 (Remarks to the Author):

The manuscript by Puan et al. demonstrates the critical effect of FUT6 on the generation of E-selectin ligands on basophils and, in the revised version, on mast cell progenitors. SNPs in the gene are demonstrated to affect FUT6 levels and the rolling interaction of basophils with endothelium in a general population cohort. Furthermore, due to random mono-allelic expression of FUT6, distinct populations of CD15s+ and CD15s- basophils and mast cell progenitors can be observed in individuals with only a single FUT6 null allele. The correlation of FUT6 null alleles with allergy- and itch-related measures and the likely connection with basophil and mast cell progenitor extravasation are important and novel contributions to the field. The authors have addressed my concerns with the original manuscript, and I have no further criticisms to offer.

Reviewer #3 (Remarks to the Author):

The authors have considered and addressed all the concerns raised by this reviewer.

We would like to thank all the Reviewers who reviewed our paper.